# RS-Del: Edit Distance Robustness Certificates for Sequence Classifiers via Randomized Deletion

**Zhuoqun Huang**
University of Melbourne
zhuoqun@unimelb.edu.au

**Neil G. Marchant**
University of Melbourne
nmarchant@unimelb.edu.au

**Keane Lucas**
Carnegie Mellon University
keanelucas@cmu.edu

**Lujo Bauer**
Carnegie Mellon University
lbauer@cmu.edu

**Olga Ohrimenko**
University of Melbourne
oohrimenko@unimelb.edu.au

**Benjamin I. P. Rubinstein**
University of Melbourne
brubinstein@unimelb.edu.au

## Abstract

Randomized smoothing is a leading approach for constructing classifiers that are certifiably robust against adversarial examples. Existing work on randomized smoothing has focused on classifiers with continuous inputs, such as images, where $\ell_p$-norm bounded adversaries are commonly studied. However, there has been limited work for classifiers with discrete or variable-size inputs, such as for source code, which require different threat models and smoothing mechanisms. In this work, we adapt randomized smoothing for discrete sequence classifiers to provide certified robustness against edit distance-bounded adversaries. Our proposed smoothing mechanism *randomized deletion* (RS-Del) applies random deletion edits, which are (perhaps surprisingly) sufficient to confer robustness against adversarial deletion, insertion and substitution edits. Our proof of certification deviates from the established Neyman-Pearson approach, which is intractable in our setting, and is instead organized around longest common subsequences. We present a case study on malware detection—a binary classification problem on byte sequences where classifier evasion is a well-established threat model. When applied to the popular MalConv malware detection model, our smoothing mechanism RS-Del achieves a certified accuracy of 91% at an edit distance radius of 128 bytes.

## 1 Introduction

Neural networks have achieved exceptional performance for classification tasks in unstructured domains, such as computer vision and natural language. However, they are susceptible to adversarial examples—inconspicuous perturbations that cause inputs to be misclassified [1, 2]. While a multitude of defenses have been proposed against adversarial examples, they have historically been broken by stronger attacks. For instance, six of nine defense papers accepted for presentation at ICLR2018 were defeated months before the conference took place [3]; another tranche of thirteen defenses were circumvented shortly after [4]. This arms race between attackers and defenders has led the community to investigate certified robustness, which aims to guarantee that a classifier's prediction cannot be changed under a specified set of adversarial perturbations [5, 6].

37th Conference on Neural Information Processing Systems (NeurIPS 2023).

Most prior work on certified robustness has focused on classifiers with *continuous fixed-dimensional* inputs, under the assumption of $\ell_p$-norm bounded perturbations. A variety of certification methods have been proposed for this setting, including deterministic methods which bound a network's output by convex relaxation [5, 7, 6, 8] or composing layerwise bounds [9, 8, 10–12], and randomized smoothing which obtains high probability bounds for a base classifier smoothed under random noise [13–15]. Randomized smoothing has achieved state-of-the-art certified robustness against $\ell_2$-norm bounded perturbations on ImageNet, correctly classifying 71% of a test set under perturbations with $\ell_2$-norm up to $127/255$ (half maximum pixel intensity) [16]. However, real-world attacks go beyond continuous modalities: a range of attacks have been demonstrated against models with discrete inputs, such as binary executables [17–20], source code [21], and PDF files [22].

Unfortunately, there has been limited investigation of certified robustness for discrete modalities, where prior work has focused on perturbations with bounded Hamming ($\ell_0$) distance [23–25], in some cases under additional constraints [26–30]. While this work covers attackers that overwrite content, it does not cover attackers that insert/delete content, which is important for instance in the malware domain [17, 31]. One exception is Zhang et al.'s method for certifying robustness under string transformation rules that may include insertion and deletion, however their approach is only feasible for small rule sets and is limited to LSTMs [32]. Liu et al. [33] also study certification for a threat model that includes insertion/deletion in the context of point cloud classification.

In this paper we develop a comprehensive treatment of *edit distance certifications* for sequence classifiers. We consider input sequences of varying length on finite domains. We cover threat models where an adversary can arbitrarily perturb sequences by substitutions, insertions, and deletions or any subset of these operations. Moreover, our framework encompasses adversaries that apply edits to blocks of tokens at a time. Such threat models are motivated in malware analysis, where attackers are likely to edit semantic objects in an executable (e.g., whole instructions), rather than at the byte-level. We introduce tunable decision thresholds, to adjust decision boundaries while still forming sound certifications. This permits trading-off misclassification rates and certification radii between classes, and is useful in settings where adversaries have a strong incentive to misclassify malicious instances as benign [34, 35].

We accomplish our certifications using randomized smoothing [13, 14], which we instantiate with a general-purpose deletion smoothing mechanism called RS-Del. Perhaps surprisingly, RS-Del does not need to sample all edit operations covered by our certifications. By smoothing using deletions only, we simplify our certification analysis and achieve the added benefit of improved efficiency by querying the base classifier with shorter sequences. RS-Del is compatible with arbitrary base classifiers, requiring only oracle query access such as via an inference API. To prove our robustness certificates, we have to deviate from the standard Neyman-Pearson approach to randomized smoothing, as it is intractable in our setting. Instead, we organize our proof around a representative longest common subsequence (LCS): the LCS serves as a reference point in the smoothed space between an input instance and a neighboring instance, allowing us to bound the confidence of the smoothed model between the two.

Finally, we present a comprehensive evaluation of RS-Del in the malware analysis setting[1]. We investigate tradeoffs between accuracy and the size of our certificates for varying levels of deletion, and observe that RS-Del can achieve a certified accuracy of 91% at an edit distance radius of 128 bytes using on the order of $10^3$ model queries. By comparison, a brute force certification would require in excess of $10^{308}$ model queries to certify the same radius. We also demonstrate asymmetric certificates (favouring the malicious class) and certificates covering edits at the machine instruction level by leveraging chunking and information from a disassembler. Finally, we assess the empirical robustness of RS-Del to several attacks from the literature, where we observe a reduction in attack success rates.

## 2   Preliminaries

**Sequence classification**   Let $\mathcal{X} = \Omega^\star$ represent the space of finite-length sequences (including the empty sequence) whose elements are drawn from a finite set $\Omega$. For a sequence $\boldsymbol{x} \in \mathcal{X}$, we denote its length by $|\boldsymbol{x}|$ and the element at position $i$ by $x_i$ where $i$ runs from 1 to $|\boldsymbol{x}|$. We consider classifiers that map sequences in $\mathcal{X}$ to $K$ classes in $\mathcal{Y} = \{0, \ldots, K-1\}$. For example, in our case study on

---

[1]Our implementation is available at `https://github.com/Dovermore/randomized-deletion`.

malware detection, we take $\mathcal{X}$ to be the space of byte sequences (binaries), and $\Omega$ to be the set of bytes and $\mathcal{Y}$ to be $\{0, 1\}$ where 0 and 1 denote benign and malicious binaries respectively.

**Robustness certification**    Given a classifier $f \colon \mathcal{X} \to \mathcal{Y}$, an input $\boldsymbol{x} \in \mathcal{X}$ and a neighborhood $\mathcal{N}(\boldsymbol{x}) \subset \mathcal{X}$ around $\boldsymbol{x}$, a *robustness certificate* is a guarantee that the classifier's prediction is constant in the neighborhood, i.e.,

$$f(\boldsymbol{x}) = f(\boldsymbol{x}') \quad \forall \boldsymbol{x}' \in \mathcal{N}(\boldsymbol{x}). \tag{1}$$

When the neighborhood corresponds to a ball of radius $r$ centered on $\boldsymbol{x}$, we make the dependence on $r$ explicit by writing $\mathcal{N}_r(\boldsymbol{x})$. We adopt the paradigm of *conservative*, *probabilistic* certification, which is a natural fit for randomized smoothing [14]. Under this paradigm, a certifier may either assert that (1) holds with high probability, or decline to assert whether (1) holds or not.

**Edit distance robustness**    Most existing work on robustness certification focuses on classifiers that operate on *fixed-dimensional* inputs in $\mathbb{R}^d$, where the neighborhood of certification is an $\ell_p$-ball [6, 5, 13–15]. However, $\ell_p$ robustness is not well motivated for sequence classifiers: $\ell_p$ neighborhoods are limited to constant-length sequences, and the norm is ill-defined for sequences with non-numeric elements. For example, an $\ell_p$ neighborhood around a byte sequence like $\boldsymbol{x} = (78, 7A, 2D, 00)$ must include sequences additively perturbed by real-valued sequences like $\boldsymbol{\delta} = (-0.09, 0.07, 0.01, 0.1)$, which clearly results in a type mismatch. Even if one focuses on robustness for length-preserving sequence perturbations, the $\ell_p$ neighborhood is a poor choice because it excludes sequences that are even slightly misaligned. Motivated by these shortcomings, we consider *edit distance* robustness.

Edit distance is a natural measure for comparing sequences. Given a set of elementary edit operations (ops) $O$, the edit distance $\mathrm{dist}_O(\boldsymbol{x}, \boldsymbol{x}')$ is the minimum number of ops required to transform sequence $\boldsymbol{x}$ into $\boldsymbol{x}'$. We consider three ops: delete a single element (del), insert a single element (ins), and substitute one element with another (sub). For generality, we allow $O$ to be any combination of these ops. When $O = \{\mathsf{del}, \mathsf{ins}, \mathsf{sub}\}$ the edit distance is known as Levenshtein distance [36]. A primary goal of this paper is to produce edit distance robustness certificates, where the neighborhood of certification is an edit distance ball $\mathcal{N}_r(\boldsymbol{x}) = \{\boldsymbol{x}' \in \mathcal{X} : \mathrm{dist}_O(\boldsymbol{x}', \boldsymbol{x}) \leq r\}$.

**Threat model**    We consider an adversary that has full knowledge of our base and smoothed models, source of randomness, certification scheme, and possesses unbounded computation. The attacker makes edits from $O$ up to some budget, in order to misclassify a target $\boldsymbol{x}$. In the context of our experimental case study on malware detection, edit distance is a reasonable proxy for the cost of running evasion attacks that iteratively apply localized functionality-preserving edits (e.g., [19, 37, 38, 17, 39]). Since the edit distance scales roughly linearly with the number of attack iterations, the adversary has an incentive to minimize edit distance for these attacks.

## 3    RS-Del: Randomized deletion smoothing

In this section, we propose a method for constructing sequence classifiers that are certifiably robust under bounded edit distance perturbations. Our method RS-Del extends randomized smoothing with a novel deletion mechanism and tunable decision thresholds. We review randomized smoothing in Section 3.1 and describe our deletion mechanism in Section 3.2, along with its practical aspects in Section 3.3. We summarize the certified robustness guarantees of our method in Table 1 and defer their derivation to Section 4.

### 3.1    Randomized smoothing

Let $f_\mathrm{b} \colon \mathcal{X} \to \mathcal{Y}$ be a *base* classifier and $\phi \colon \mathcal{X} \to \mathcal{D}(\mathcal{X})$ be a *smoothing mechanism* that maps inputs to a distributions over (perturbed) inputs. Randomized smoothing composes $f_\mathrm{b}$ and $\phi$ to construct a new *smoothed* classifier $f \colon \mathcal{X} \to \mathcal{Y}$. For any input $\boldsymbol{x} \in \mathcal{X}$, the smoothed classifier's prediction is

$$f(\boldsymbol{x}) := \underset{y \in \mathcal{Y}}{\arg\max} \left\{ p_y(\boldsymbol{x}; f_\mathrm{b}, \phi) - \eta_y \right\}, \tag{2}$$

where $\boldsymbol{\eta} = \{\eta_y\}_{y \in \mathcal{Y}}$ is a set of real-valued tunable decision thresholds and

$$p_y(\boldsymbol{x}; f_\mathrm{b}, \phi) = \Pr_{\boldsymbol{z} \sim \phi(\boldsymbol{x})} [f_\mathrm{b}(\boldsymbol{z}) = y] \tag{3}$$

is the probability that the base classifier $f_{\rm b}$ predicts class $y$ for a perturbed input drawn from $\phi(\boldsymbol{x})$. We omit the dependence of $p_y$ on $f_{\rm b}$ and $\phi$ where it is clear from context.

The viability of randomized smoothing as a method for achieving certified robustness is strongly dependent on the smoothing mechanism. Ideally, the mechanism should be chosen to yield a smoothed classifier with improved robustness under the chosen threat model, while minimizing any drop in accuracy compared to the base classifier. The mechanism should also be amenable to analysis, so that a tractable robustness certificate can be derived.

*Remark* 1. Previous definitions of randomized smoothing (e.g., [13, 14]) do not incorporate decision thresholds, and effectively assume $\boldsymbol{\eta} = 0$. We introduce decision thresholds as a way to trade off error rates and robustness between classes. This is useful when there is asymmetry in misclassification costs across classes. For instance, in our case study on malware detection, robustness of benign examples is less important because adveraries have limited incentive to trigger misclassification of benign examples [34, 35, 40]. We note that the base classifier may also be equipped with decision thresholds, which provide another degree of freedom to trade off error rates between classes.

### 3.2 Randomized deletion mechanism

We propose a smoothing mechanism that achieves certified edit distance robustness. Our smoothing mechanism perturbs a sequence by deleting elements at random, and is called *randomized deletion*, or RS-Del for short.

Consider a sequence $\boldsymbol{x} \in \mathcal{X}$ whose elements are indexed by the set $[\boldsymbol{x}] = \{1, \ldots, |\boldsymbol{x}|\}$. We specify the distribution of $\phi(\boldsymbol{x})$ for RS-Del in two stages. In the first stage, a random edit $\epsilon$ is drawn from a distribution $G(\boldsymbol{x})$ over the space of possible edits to $\boldsymbol{x}$, denoted $\mathcal{E}(\boldsymbol{x})$. Since we only consider deletions for smoothing, any edit can be represented by the set of element indices in $[\boldsymbol{x}]$ that *remain* after deletion. Hence $\mathcal{E}(\boldsymbol{x})$ is taken as the powerset of $[\boldsymbol{x}]$. We specify edit distribution $G(\boldsymbol{x})$ so that each element is deleted i.i.d. with probability $p_{\rm del} \in (0, 1)$:

$$\Pr[G(\boldsymbol{x}) = \epsilon] = \prod_{i=1}^{|\boldsymbol{x}|} p_{\rm del}^{\mathbf{1}_{i \notin \epsilon}} (1 - p_{\rm del})^{\mathbf{1}_{i \in \epsilon}}, \tag{4}$$

where $\mathbf{1}_A$ denotes the indicator function, which returns 1 if $A$ evaluates to true and 0 otherwise. In the second stage, the edit $\epsilon$ is applied to $\boldsymbol{x}$ to yield the perturbed sequence:

$$\boldsymbol{z} = \mathrm{apply}(\boldsymbol{x}, \epsilon) \coloneqq \left( x_{\epsilon_{(i)}} \right)_{i=1 \ldots |\epsilon|}, \tag{5}$$

where $\epsilon_{(i)}$ denotes the $i$-th smallest index in $\epsilon$. The perturbed sequence $\boldsymbol{z}$ is guaranteed to be a subsequence of $\boldsymbol{x}$. Putting both stages together, the distribution of $\phi(\boldsymbol{x})$ is

$$\Pr[\phi(\boldsymbol{x}) = \boldsymbol{z}] = \sum_{\epsilon \in \mathcal{E}(\boldsymbol{x})} \Pr[G(\boldsymbol{x}) = \epsilon] \mathbf{1}_{\mathrm{apply}(\boldsymbol{x}, \epsilon) = \boldsymbol{z}}. \tag{6}$$

*Remark* 2. It may be surprising that we are proposing a smoothing mechanism for certified edit distance robustness that does not use the full set of edit ops $O$ covered by the threat model. It is a misconception that randomized smoothing requires perfect alignment between the mechanism and the threat model. All that is needed from a robustness perspective, is for the mechanism to return distributions that are statistically close for any pair of inputs that are close in $O$ edit distance; this can be achieved solely with deletion. In fact, perfect alignment is known to be suboptimal for some $\ell_p$ threat models [15]. Our deletion mechanism leads to a tractable robustness certificate covering the full set of edit ops (see Section 4). Moreover while benefiting robustness, our empirical results show that our deletion mechanism has only a minor impact on accuracy (see Section 5). Finally, our deletion mechanism reduces the length of the input, which is beneficial for computational efficiency (see Appendix G). This is not true in general for mechanisms employing insertions/substitutions.

### 3.3 Practical considerations

We now discuss considerations for implementing and certifying RS-Del in practice. In doing so, we reference theoretical results for certification, which are covered later in Section 4.

1   Sample perturbed inputs for prediction estimate: $S \leftarrow \{z_i \sim \phi(x)\}_{i=1...n_{\text{pred}}}$
2   Confidence scores: $\hat{\mu}_y \leftarrow \frac{1}{|S|} \sum_{z \in S} \mathbf{1}_{f_b(z)=y}$
3   Prediction: $\hat{y} \leftarrow \arg\max_{y \in \mathcal{Y}} \{\hat{\mu}_y - \eta_y\}$
4   Sample perturbed inputs for lower confidence bound (LCB): $S' \leftarrow \{z_i \sim \phi(x)\}_{i=1...n_{\text{bnd}}}$
5   LCB: $\underline{\mu}_{\hat{y}} \leftarrow \texttt{BinLCB}(\sum_{z \in S'} \mathbf{1}_{f_b(z)=\hat{y}}, |S'|, \alpha)$
6   Compute certified radius $r^\star$ using Table 1
7   **if** $\underline{\mu}_{\hat{y}} < \eta_{\hat{y}}$ **then** **return** *abstain*
8   **else** **return** *prediction $\hat{y}$, radius $r^\star$*

Figure 1: Probabilistic certification of RS-Del. Here $x$ is the input sequence, $f_b$ is the base classifier, $p_{\text{del}}$ is the deletion probability, $\eta$ is the set of decision thresholds, $\alpha$ is the significance level, and $n_{\text{pred}}, n_{\text{bnd}}$ are sample sizes. $\texttt{BinLCB}(k, n, \alpha)$ returns a lower confidence bound for $p$ at level $\alpha$ given $k \sim \text{Bin}(n, p)$.

| Edit ops $O$ | | | |
| --- | --- | --- | --- |
| del | ins | sub | Certified radius $r^\star$ |
| | ✓ | | $\left\lfloor \log\frac{1-\mu_y}{1-\nu_y} / \log p_{\text{del}} \right\rfloor$ |
| ✓ | | | $\left\lfloor \log\frac{\nu_y}{\mu_y} / \log p_{\text{del}} \right\rfloor$ |
| ✓ | ✓ | | $\left\lfloor \log\frac{\nu_y}{\mu_y} / \log p_{\text{del}} \right\rfloor$ |
| ✓ | ✓ | ✓ | $\left\lfloor \log(1+\nu_y-\mu_y) / \log p_{\text{del}} \right\rfloor$ |
| | | ✓ | $\left\lfloor \log(1+\nu_y-\mu_y) / \log p_{\text{del}} \right\rfloor$ |
| | ✓ | ✓ | $\left\lfloor \log(1+\nu_y-\mu_y) / \log p_{\text{del}} \right\rfloor$ |
| ✓ | | ✓ | $\left\lfloor \log(1+\nu_y-\mu_y) / \log p_{\text{del}} \right\rfloor$ |

Table 1: Edit distance robustness certificates for RS-Del as a function of the edit ops $O$ used to define the edit distance. Here $\mu_y$ is the confidence for predicted class $y$ and $\nu_y$ is a threshold derived from $\eta$ defined in (10).

**Probabilistic certification**   Randomized smoothing does not generally support exact evaluation of the classifier's confidence $\mu_y := p_y(x)$, which is required for exact prediction and exact evaluation of the certificates we develop in Section 4. While $\mu_y$ can be evaluated exactly for RS-Del by enumerating over the possible edits $\mathcal{E}(x)$, the computation scales exponentially in $|x|$ (see Appendix A). Since this is infeasible for even moderately-sized $x$, we follow standard practice in randomized smoothing and estimate $\mu_y$ with a lower confidence bound using Monte Carlo sampling [14]. This procedure is described in pseudocode in Figure 1: lines 1–3 estimate the predicted class $y = f(x)$, lines 4–5 compute a $1 - \alpha$ lower confidence bound on $\mu_y$, and line 6 uses this information and the results in Table 1 to compute a probabilistic certificate that holds with probability $1 - \alpha$. If the lower confidence bound on $\mu_y$ exceeds the corresponding detection threshold $\eta_y$, the prediction and certificate are returned (line 8), otherwise we *abstain* due to lack of statistical significance (line 7).

**Training**   While randomized smoothing is compatible with any base classifier, it generally performs poorly for conventionally-trained classifiers [13]. We therefore train base classifiers specifically to be used with RS-Del, by replacing original sequences with perturbed sequences (drawn from $\phi$) at training time. This has been shown to achieve good empirical performance in prior work [14].

**Sequence chunking**   So far, we have described edit distance robustness and RS-Del assuming edits are applied at the level of sequence elements. However in some applications it may be reasonable to assume edits are applied at a coarser level, to contiguous chunks of sequence elements. For example, in malware analysis, one can leverage information from a disassembler to group low-level sequence elements (bytes) into more semantically meaningful chunks, such as machine instructions, addresses and header fields (see Appendix C.3). Our methods are compatible with chunking—we simply reinterpret the sequence as a sequence of chunks, rather than a sequence of lower-level elements. This can yield a tighter robustness certificate, since edits within chunks are excluded.

## 4   Edit distance robustness certificate

We now derive an edit distance robustness certificate for RS-Del. We present the derivation in three parts: Section 4.1 provides an outline, Section 4.2 derives a lower bound on RS-Del's confidence score and Section 4.3 uses the bound to complete the derivation. All proofs are presented in Appendix B.

### 4.1   Derivation outline

Following prior work [14, 13, 41], we derive an edit distance robustness certificate that relies on limited information about RS-Del. We allow the certificate to depend on the input $x$, the smoothed prediction $y = f(x)$, the confidence score $\mu_y = p_y(x; f_b)$, the decision threshold $\eta_y$, and the architecture of $f$, including the deletion smoothing mechanism $\phi$, but excluding the architecture of

the base classifier $f_b$. Limiting the dependence in this way improves tractability and ensures that the certificate is applicable for any choice of base classifier $f_b$. Formally, the only information we assume about $f_b$ is that it is some classifier in the feasible base classifier set:

$$\mathcal{F}(\boldsymbol{x}, \mu_y) = \{ h \in \mathcal{X} \to \mathcal{Y} : \mu_y = p_y(\boldsymbol{x}; h) \} . \tag{7}$$

Recall that an edit distance robustness certificate at radius $r$ for a classifier $f$ at input $\boldsymbol{x}$ is a guarantee that $f(\boldsymbol{x}) = f(\bar{\boldsymbol{x}})$ for any perturbed input $\bar{\boldsymbol{x}}$ in the neighborhood $\mathcal{N}_r(\boldsymbol{x}) = \{\bar{\boldsymbol{x}} \in \mathcal{X} : \mathrm{dist}_O(\bar{\boldsymbol{x}}, \boldsymbol{x}) \leq r\}$. We observe that this guarantee holds for RS-Del in the limited information setting iff the $\boldsymbol{\eta}$-adjusted confidence for predicted class $y$ exceeds the $\boldsymbol{\eta}$-adjusted confidence for any other class for all perturbed inputs $\bar{\boldsymbol{x}}$ and feasible base classifiers $h$:

$$p_y(\bar{\boldsymbol{x}}; h) - \eta_y \geq \max_{y' \neq y}\{p_{y'}(\bar{\boldsymbol{x}}; h) - \eta_{y'}\} \quad \forall \bar{\boldsymbol{x}} \in \mathcal{N}_r(\boldsymbol{x}), h \in \mathcal{F}(\boldsymbol{x}, \mu_y). \tag{8}$$

To avoid dependence on the confidence of the runner-up class, which is inefficient for probabilistic certification, we work with the following more convenient condition.

**Proposition 3.** *A sufficient condition for* (8) *is* $\rho(\boldsymbol{x}, \mu_y) \geq \nu_y(\boldsymbol{\eta})$ *where*

$$\rho(\boldsymbol{x}, \mu_y) \coloneqq \min_{\bar{\boldsymbol{x}} \in \mathcal{N}_r(\boldsymbol{x})} \min_{h \in \mathcal{F}(\boldsymbol{x}, \mu_y)} p_y(\bar{\boldsymbol{x}}; h) \tag{9}$$

*is a tight lower bound on the confidence for class $y$, and we define the threshold*

$$\nu_y(\boldsymbol{\eta}) = \begin{cases} \frac{1}{2} + \eta_y - \min_{y' \neq y} \eta_{y'}, & \eta_y \geq \min_{y' \neq y} \eta_{y'} \text{ and } |\mathcal{Y}| > 2, \\ 1 + \eta_y - \min_{y' \neq y} \eta_{y'}, & \eta_y < \min_{y' \neq y} \eta_{y'} \text{ and } |\mathcal{Y}| > 2, \\ \frac{1 + \eta_y - \min_{y' \neq y} \eta_{y'}}{2}, & |\mathcal{Y}| = 2. \end{cases} \tag{10}$$

The standard approach for evaluating $\rho(\boldsymbol{x}, \mu_y)$ is via the Neyman-Pearson lemma [42, 14], however this seems insurmountable in our setting due to the challenging geometry of the edit distance neighborhood. We therefore proceed by deriving a loose lower bound on the confidence $\tilde{\rho}(\boldsymbol{x}, \mu_y) \leq \rho(\boldsymbol{x}, \mu_y)$, noting that the robustness guarantee still holds so long as $\tilde{\rho}(\boldsymbol{x}, \mu_y) > \nu_y(\boldsymbol{\eta})$. The derivation proceeds in two steps. In the first step, covered in Section 4.2, we derive a lower bound for the inner minimization in (9), which we denote by $\tilde{\rho}(\bar{\boldsymbol{x}}, \boldsymbol{x}, \mu_y)$. Then in the second step, covered in Section 4.3, we complete the derivation by minimizing $\tilde{\rho}(\bar{\boldsymbol{x}}, \boldsymbol{x}, \mu_y)$ over the edit distance neighborhood. Our results are summarized in Table 1, where we provide certificates under various constraints on the edit ops.

## 4.2 Minimizing over feasible base classifiers

In this section, we derive a loose lower bound on the classifier confidence with respect to feasible base classifiers

$$\tilde{\rho}(\bar{\boldsymbol{x}}, \boldsymbol{x}, \mu_y) \leq \rho(\bar{\boldsymbol{x}}, \boldsymbol{x}, \mu_y) = \min_{h \in \mathcal{F}(\boldsymbol{x}, \mu_y)} p_y(\bar{\boldsymbol{x}}; h). \tag{11}$$

To begin, we write $p_y(\bar{\boldsymbol{x}}; h)$ as a sum over the edit space by combining (3) and (6):

$$p_y(\bar{\boldsymbol{x}}; h) = \sum_{\bar{\epsilon} \in \mathcal{E}(\bar{\boldsymbol{x}})} s(\bar{\epsilon}, \bar{\boldsymbol{x}}; h) \quad \text{where} \quad s(\bar{\epsilon}, \bar{\boldsymbol{x}}; h) = \Pr\left[G(\bar{\boldsymbol{x}}) = \bar{\epsilon}\right] \mathbf{1}_{h(\mathrm{apply}(\bar{\boldsymbol{x}}, \bar{\epsilon})) = y}. \tag{12}$$

We would like to rewrite this sum in terms of the known confidence score at $\boldsymbol{x}$, $\mu_y = p_y(\boldsymbol{x}; h) = \sum_{\epsilon \in \mathcal{E}(\boldsymbol{x})} s(\epsilon, \boldsymbol{x}; h)$. To do so, we identify pairs of edits $\bar{\epsilon}$ to $\bar{\boldsymbol{x}}$ and $\epsilon$ to $\boldsymbol{x}$ for which the corresponding terms $s(\bar{\epsilon}, \bar{\boldsymbol{x}}; h)$ and $s(\epsilon, \boldsymbol{x}; h)$ are proportional.

**Lemma 4** (Equivalent edits). *Let $\boldsymbol{z}^\star$ be a longest common subsequence (LCS) [43] of $\bar{\boldsymbol{x}}$ and $\boldsymbol{x}$, and let $\bar{\epsilon}^\star \in \mathcal{E}(\bar{\boldsymbol{x}})$ and $\epsilon^\star \in \mathcal{E}(\boldsymbol{x})$ be any edits such that $\mathrm{apply}(\bar{\boldsymbol{x}}, \bar{\epsilon}^\star) = \mathrm{apply}(\boldsymbol{x}, \epsilon^\star) = \boldsymbol{z}^\star$. Then there exists a bijection $m : 2^{\bar{\epsilon}^\star} \to 2^{\epsilon^\star}$ such that $\mathrm{apply}(\bar{\boldsymbol{x}}, \bar{\epsilon}) = \mathrm{apply}(\boldsymbol{x}, \epsilon)$ for any $\bar{\epsilon} \subseteq \bar{\epsilon}^\star$ and $\epsilon = m(\bar{\epsilon})$. Furthermore, we have $s(\bar{\epsilon}, \bar{\boldsymbol{x}}; h) = p_{\mathrm{del}}^{|\bar{\boldsymbol{x}}| - |\boldsymbol{x}|} s(\epsilon, \boldsymbol{x}; h)$.*

Applying this proportionality result to all pairs of edits $\bar{\epsilon}, \epsilon$ related under the bijection $m$ yields:

$$\sum_{\bar{\epsilon} \in 2^{\bar{\epsilon}^\star}} s(\bar{\epsilon}, \bar{\boldsymbol{x}}; h) = p_{\mathrm{del}}^{|\bar{\boldsymbol{x}}| - |\boldsymbol{x}|} \sum_{\epsilon \in 2^{\epsilon^\star}} s(\epsilon, \boldsymbol{x}; h).$$

Thus we can achieve our goal of writing $p_y(\bar{\boldsymbol{x}}; h)$ in terms of $\mu_y$. A rearrangement of terms gives:

$$p_y(\bar{\boldsymbol{x}}; h) = p_{\mathsf{del}}^{|\bar{\boldsymbol{x}}|-|\boldsymbol{x}|} \left( \mu_y - \sum_{\epsilon \notin 2^{\epsilon^\star}} s(\epsilon, \boldsymbol{x}; h) \right) + \sum_{\bar{\epsilon} \notin 2^{\bar{\epsilon}^\star}} s(\bar{\epsilon}, \bar{\boldsymbol{x}}; h). \tag{13}$$

This representation is convenient for deriving a lower bound. Specifically, we can drop the sum over $\bar{\epsilon} \notin 2^{\bar{\epsilon}^\star}$ and upper-bound the sum over $\epsilon \notin 2^{\epsilon^\star}$ to obtain a lower bound that is independent of $h$.

**Theorem 5.** *For any pair of inputs $\bar{\boldsymbol{x}}, \boldsymbol{x} \in \mathcal{X}$ we have*

$$\rho(\bar{\boldsymbol{x}}, \boldsymbol{x}, \mu_y) \geq \tilde{\rho}(\bar{\boldsymbol{x}}, \boldsymbol{x}, \mu_y) = p_{\mathsf{del}}^{|\bar{\boldsymbol{x}}|-|\boldsymbol{x}|} \left( \mu_y - 1 + p_{\mathsf{del}}^{\frac{1}{2}(\mathrm{dist}_{\mathrm{LCS}}(\bar{\boldsymbol{x}}, \boldsymbol{x})+|\boldsymbol{x}|-|\bar{\boldsymbol{x}}|)} \right). \tag{14}$$

*where $\mathrm{dist}_{\mathrm{LCS}}(\bar{\boldsymbol{x}}, \boldsymbol{x})$ is the longest common subsequence (LCS) distance[2] between $\bar{\boldsymbol{x}}$ and $\boldsymbol{x}$.*

### 4.3 Minimizing over the edit distance neighborhood

In this section, we complete the derivation of our robustness certificate by minimizing the lower bound in (14) over the edit distance neighborhood:

$$\tilde{\rho}(\boldsymbol{x}; \mu_y) = \min_{\bar{\boldsymbol{x}} \in \mathcal{N}_r(\boldsymbol{x})} \tilde{\rho}(\bar{\boldsymbol{x}}, \boldsymbol{x}, \mu_y). \tag{15}$$

We are interested in general edit distance neighborhoods, where the edit ops $O$ used to define the edit distance may be constrained. For example, if the attacker is capable of performing elementary substitutions and insertions, but not deletions, then $O = \{\mathsf{sub}, \mathsf{ins}\}$. As a step towards solving (15), it is therefore useful to express $\tilde{\rho}(\bar{\boldsymbol{x}}, \boldsymbol{x}, \mu_y)$ in terms of edit op counts, as shown below.

**Corollary 6.** *Suppose there exists a sequence of edits from $\bar{\boldsymbol{x}}$ to $\boldsymbol{x}$ that consists of $n_{\mathsf{sub}}$ substitutions, $n_{\mathsf{ins}}$ insertions and $n_{\mathsf{del}}$ deletions s.t. $n_{\mathsf{sub}} + n_{\mathsf{ins}} + n_{\mathsf{del}} = \mathrm{dist}_O(\bar{\boldsymbol{x}}, \boldsymbol{x})$ and $n_{\mathsf{sub}}, n_{\mathsf{ins}}, n_{\mathsf{del}} \geq 0$. Then*

$$\tilde{\rho}(\bar{\boldsymbol{x}}, \boldsymbol{x}, \mu_y) = p_{\mathsf{del}}^{n_{\mathsf{del}}-n_{\mathsf{ins}}} \left( \mu_y - 1 + p_{\mathsf{del}}^{n_{\mathsf{sub}}+n_{\mathsf{ins}}} \right).$$

This parameterization of the lower bound enables us to re-express (15) as an optimization problem over edit ops counts:

$$\tilde{\rho}(\boldsymbol{x}; \mu_y) = \min_{n_{\mathsf{sub}}, n_{\mathsf{ins}}, n_{\mathsf{del}} \in \mathcal{C}_r} p_{\mathsf{del}}^{n_{\mathsf{del}}-n_{\mathsf{ins}}} \left( \mu_y - 1 + p_{\mathsf{del}}^{n_{\mathsf{sub}}+n_{\mathsf{ins}}} \right), \tag{16}$$

where $\mathcal{C}_r$ encodes constraints on the set of counts. If any number of insertions, deletions or substitutions are allowed, then the edit distance is known as the *Levenshtein distance* and $\mathcal{C}_r$ consists of sets of counts that sum to $r$. We solve the minimization problem for this case below.

**Theorem 7** (Levenshtein distance certificate). *A lower bound on the classifier's confidence within the Levenshtein distance neighborhood $\mathcal{N}_r(\boldsymbol{x})$ (with $O = \{\mathsf{del}, \mathsf{ins}, \mathsf{sub}\}$) is $\tilde{\rho}(\boldsymbol{x}; \mu_y) = \mu_y - 1 + p_{\mathsf{del}}^r$. It follows that the classifier is certifiably robust for any Levenshtein distance ball with radius $r$ less than or equal to the certified radius $r^\star = \lfloor \log(1+\nu_y(\boldsymbol{\eta})-\mu_y)/\log p_{\mathsf{del}} \rfloor$.*

It is straightforward to adapt this result to account for constraints on the edit ops $O$. Results for all combinations of edit ops are provided in Table 1.

So far in this section we have obtained results that depend on the classifier's confidence $\mu_y$, assuming it can be evaluated exactly. However since exact evaluation of $\mu_y$ is not feasible in general (see Section 3.3), we extend our results to the probabilistic setting, assuming a $1 - \alpha$ lower confidence bound on $\mu_y$ is available. This covers the probabilistic certification procedure described in Figure 1.

**Corollary 8.** *Suppose the procedure in Figure 1 returns predicted class $\hat{y}$ with certified radius $r^\star$. Then an edit distance robustness certificate of radius $r \leq r^\star$ holds at $\boldsymbol{x}$ with probability $1 - \alpha$.*

## 5 Case study: robust malware detection

We now present a case study on the application of RS-Del to malware detection. We report on certified accuracy for Levenshtein and Hamming distance threat models. We show that by tuning decision thresholds we can increase certified radii for the malicious class while maintaining accuracy. We also evaluate RS-Del on a range of published malware classifier attacks. Due to space constraints, we present the complete study in Appendices C–F, where we report on training curves, the computational cost of training and certification, certified radii normalized by file size, and results where byte edits are chunked by instructions.

---

[2]The LCS distance is equivalent to the generalized edit distance with $O = \{\mathsf{del}, \mathsf{ins}\}$.

**Background**   Malware (malicious software) detection is a long standing problem in security where machine learning is playing an increasing role [44–47]. Inspired by the success of neural networks in other domains, recent work has sought to design neural network models for static malware detection which operate on raw binary executables, represented as byte sequences [48–50]. While these models have achieved competitive performance, they are vulnerable to adversarial perturbations that allow malware to evade detection [18–20, 37, 51, 31, 52, 17, 39]. Our edit distance threat model reflects these attacks in the malware domain where, even though a variety of perturbations with different semantic effects are possible, any perturbation can be represented in terms of elementary byte deletions, insertions and substitutions. For perspective on the threat model, consider YARA [53], a rule-based tool that is widely used for static malware analysis in industry. Running Nextron System's YARA rule set[3] on a sample of binaries from the VTFeed dataset (introduced below), we find 83% of rule matches are triggered by fewer than 128 bytes. This implies most rules can be evaded by editing fewer than 128 bytes—a regime that is covered by our certificates in some instances (see Table 2). Further background and motivation for the threat model is provided in Appendix C, along with a reduction of static malware detection to sequence classification.

**Experimental setup**   We use two Windows malware datasets: Sleipnir2 which is compiled from public sources following Al-Dujaili et al. [54] and VTFeed which is collected from VirusTotal [17]. We consider three malware detection models: a model smoothed with our randomized deletion mechanism (RS-Del), a model smoothed with the randomized ablation mechanism proposed by Levine and Feizi [24] (RS-Abn), and a non-smoothed baseline (NS). All of the models are based on a popular CNN architecture called MalConv [48], and are evaluated on a held-out test set. We emphasize that RS-Del is the only model that provides edit distance certificates (general $O$), while RS-Abn provides a Hamming distance certificate ($O = \{\text{sub}\}$). We review RS-Abn in Appendix I, where we describe modifications required for discrete sequence classification, and provide an analysis comparing the Hamming distance certificates of RS-Abn and RS-Del. Details about the datasets, models, training procedure, calibration, parameter settings and hardware are provided in Appendix D.

**Accuracy/robustness tradeoffs**   Our first set of experiments investigate tradeoffs between malware detection accuracy and robustness guarantees as parameters associated with the smoothed models are varied. Table 2 reports clean accuracy, median certified radius (CR) and median certified radius normalized by file size (NCR) on the test set for RS-Del as a function of $p_{\text{del}}$. A reasonable tradeoff is observed at $p_{\text{del}} = 99.5\%$ for Sleipnir2, where a median certified radius of 137 bytes is attained with only a 2-3% drop in clean accuracy from the NS baseline. The corresponding median NCR is 0.06% and varies in the range 0–9% across the test set. We also vary the decision threshold $\eta$ at $p_{\text{del}} = 99.5\%$ for Sleipnir2, and obtain asymmetrical robustness guarantees with a median certified radius up to 582 bytes for the malicious class, while maintaining the same accuracies (see Table 8 in Appendix E.1).

Since there are no baseline methods that support edit distance certificates, we compare with RS-Abn, which produces a limited Hamming distance certificate. Figure 2 plots the certified accuracy curves for RS-Del and RS-Abn for different values of the associated smoothing parameters $p_{\text{del}}$ and $p_{\text{ab}}$. The certified accuracy is the fraction of instances in the test set for which the model's prediction is correct *and* certifiably robust at radius $r$, and is therefore sensitive to both robustness and accuracy. We find that RS-Del outperforms RS-Abn in terms of certified accuracy at all radii $r$ when $p_{\text{del}} = p_{\text{ab}}$, while covering a larger set of inputs (since RS-Del's edit distance ball includes RS-Abn's Hamming ball). Further results and interpretation for these experiments are provided in Appendix E.

**Empirical robustness to attacks**   Our edit distance certificates are conservative and may underestimate robustness to adversaries with additional constraints (e.g., maintaining executability, preserving a malicious payload, etc.). To provide a more complete picture of robustness, we subject RS-Del and NS to six recently published attacks [20, 37, 17, 38, 31] covering white-box and black-box settings. We adapt gradient-based white-box attacks [20, 17] for randomized smoothing in Appendix H. We do not constrain the number of edits each attack can make, which yields adversarial examples *well outside* the edit distance we can certify for four of six attacks (see Table 9 of Appendix F). We measure robustness in terms of the *attack success rate*, defined as the fraction of instances in the evaluation set for which the attack flips the prediction from malicious to benign on at least one repeat (we repeat each attack five times). For both Sleipnir2 and VTFeed, we observe that RS-Del achieves

---

[3]`https://valhalla.nextron-systems.com/`

| Model | $p_{del}$ | Clean Accuracy | Median CR | Median NCR % |
|---|---|---|---|---|
| Sleipnir2 dataset | | | | |
| NS | — | 98.9% | — | — |
| RS-Del | 90% | 97.1% | 6 | 0.0023 |
| | 95% | 97.8% | 13 | 0.0052 |
| | 97% | 97.4% | 22 | 0.0093 |
| | 99% | 98.1% | 68 | 0.0262 |
| | **99.5%** | **96.5%** | **137** | **0.0555** |
| | 99.9% | 83.7% | 688 | 0.2269 |
| VTFeed dataset | | | | |
| NS | — | 98.9% | — | — |
| RS-Del | 97% | 92.1% | 22 | 0.0045 |
| | 99% | 86.9% | 68 | 0.0122 |

Table 2: Clean accuracy and robustness metrics for RS-Del as a function of dataset and deletion probability $p_{del}$. All metrics are computed on the test set. "Median CR" is the median certified Levenshtein distance radius in bytes and "median NCR %" is the median certified Levenshtein distance radius normalized as a percentage of the file size. A good tradeoff is achieved when $p_{del} = 99.5\%$ (in bold).

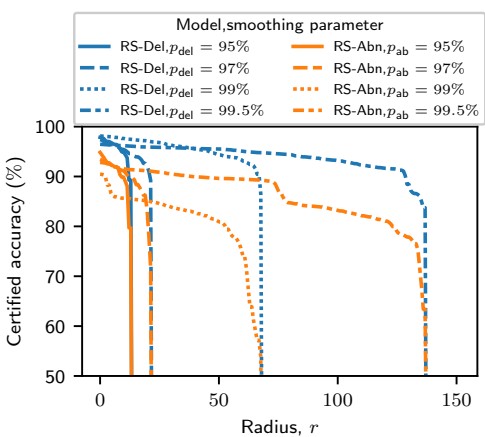

Figure 2: Certified accuracy for RS-Del and RS-Abn [24] as a function of certificate radius (horizontal axis) and strength of the smoothing parameter (line style). Results are plotted for the Sleipnir2 test set. While RS-Del provides a Levenshtein distance certificate (with $O = \{del, ins, sub\}$), RS-Abn provides a more limited Hamming distance certificate ($O = \{sub\}$). The non-smoothed, non-certified model (NS) achieves a clean accuracy of 98% in this setting.

the lowest attack success rate (best robustness) for four of six attacks. In particular, we observe a 20 percentage point decrease in the success rate of *Disp*—an attack with no known defense [17]. We refer the reader to Appendix F for details of this experiment, including setup, results and discussion.

# 6 Related work

There is a rich body of research on certifications under $\ell_p$-norm-bounded threat models [5, 7, 6, 8–11, 13, 14, 12, 55, 25, 56]. While a useful abstraction for computer vision, such certifications are inadequate for many problems including perturbations to executable files considered in this work. Even in computer vision, $\ell_p$-norm bounded defenses can be circumvented by image translation, rotation, blur, and other human-imperceptible transformations that induce extremely large $\ell_p$ distances. One solution is to re-parametrize the norm-bounded distance in terms of image transformation parameters [57–59]. NLP faces a different issue: while the $\ell_0$ threat model covers adversarial word substitution [28, 60], it is too broad and covers many natural (non-adversarial) examples as well. For example, "He loves cat" and "He hates cat" are 1 word in $\ell_0$ distance from "He likes cat", but are semantically different. A radius 1 certificate will force a wrong prediction for at least one neighbor. To address this, Jia et al. [26] and Ye et al. [27] constrain the threat model to synonyms only.

In this paper we go beyond the $\ell_0$ word substitution threat model of previous work [23–25, 56], as consideration of insertions and deletions is necessary in domains such as malware analysis. Such edits are not captured by the $\ell_0$ threat model: there is no fixed input size, and even when edits are size-preserving, a few edits may lead to large $\ell_0$ distances. Arguably, our edit distance threat model for sequences and RS-Del mechanism are of independent interest to natural language also.

Certification has been studied for variations of edit distance defined for sets and graphs. Liu et al. [33] apply randomized smoothing using a subsampling mechanism to certify point cloud classifiers against edits that include insertion, deletion and substitution of points. Since a point cloud is an *unordered* set, the edit distance takes a simpler form than for sequences—it can be expressed in terms of set cardinalities rather than as a cost minimization problem over edit paths. This simplifies the analysis, allowing Liu et al. to obtain a tight certificate via the Neyman-Pearson lemma, which is not feasible

for sequences. In parallel work, Schuchardt et al. [61] consider edit distance certification for graph classification, as an application of a broader certification framework for group equivariant tasks. They apply sparsity-aware smoothing [62] to an isomorphism equivariant base classifier, to yield a smoothed classifier that is certifiably robust under insertions/deletions of edges and node attributes.

Numerous empirical defense methods have been proposed to improve robustness of machine learning classifiers in security systems [63, 64, 29, 65, 22]. Incer Romeo et al. [34] and Chen et al. [65] develop classifiers that are verifiably robust if their manually crafted features conform to particular properties (e.g., monotonicity, stability). These approaches permit a combination of (potentially vulnerable) learned behavior with domain knowledge, and thereby aim to mitigate adversarial examples. Chen et al. [22] seek guarantees against subtree insertion and deletion for PDF malware classification. Using convex over-approximation [5, 66] previously applied to computer vision, they certify fixed-input dimension classifiers popular in PDF malware analysis. Concurrent to our work, Saha et al. [29] propose to certify classifiers against patch-based attacks by aggregating predictions of fixed-sized chunks of input binaries. The patch attack threat model, however, is not widely assumed in the evasion literature for malware detectors and can be readily broken by many published attacks [17, 20]. Moreover, their de-randomized smoothing design assumes a fixed-width input (via padding and/or trimming) and reduces patch-based attacks to gradient-based $\ell_p$ attacks. While tight analysis exists for arbitrary randomized smoothing mechanisms [23], they are computationally infeasible with the edit distance threat model. Overall, we are the first to explore certified adversarial defenses that apply to sequence classifiers under the edit distance threat model.

## 7 Conclusion

In this paper, we study certified robustness of discrete sequence classifiers. We identify critical limitations of the $\ell_p$-norm bounded threat model in sequence classification and propose edit distance-based robustness, covering substitution, deletion and insertion perturbations. We then propose a novel deletion smoothing mechanism called RS-Del that is equipped with certified guarantees under several constraints on the edit operations. We present a case study of RS-Del and its certifications applied to malware analysis. We consider two malware datasets using a recent static deep malware detector, MalConv [48]. We find that RS-Del can certify radii as large as $128$ bytes (in Levenshtein distance) without significant loss in detection accuracy. A certificate of this size covers in excess of $10^{606}$ files in the proximity of a 10KB input file (see Appendix A). Results also demonstrate RS-Del improving robustness against published attacks well beyond the certified radius.

**Broader impact and limitations**  Robustness certification seeks to quantify the risk of adversarial examples while randomized smoothing both enables certification and acts to mitigate the impact of attacks. Randomized smoothing can degrade (benign) accuracy of undefended models as demonstrated in our results at higher smoothing levels. While we have strived to select high quality datasets for our case study, we note that accuracy-robustness tradeoffs may vary for different datasets and/or model architectures. Our approach is scalable relative to alternative certification strategies, however it does incur computational overheads. Finally, it is known that randomized smoothing can have disparate impacts on class-wise accuracy [67].

## Acknowledgments

This work was supported by the Department of Industry, Science, and Resources, Australia under AUSMURI CATCH, and the U.S. Army Research Office under MURI Grant W911NF2110317.

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

## A  Brute-force edit distance certification

In this appendix, we show that an edit distance certification mechanism based on brute-force search is computationally infeasible. Suppose we are interested in issuing an edit distance certificate at radius $r$ for a sequence classifier $f$ at input $\boldsymbol{x}$. Recall from (1) that in order to issue a certificate, we must show there exists no input $\bar{\boldsymbol{x}}$ within the edit distance neighborhood $\mathcal{N}_r(\boldsymbol{x})$ that would change $f$'s prediction. This problem can theoretically be tackled in a brute-force manner, by querying $f$ for all inputs in $\mathcal{N}_r(\boldsymbol{x})$. In the best case, this would take time linear in $|\mathcal{N}_r(\boldsymbol{x})|$, assuming $f$ responds to queries in constant time. However the following lower bound [68], shows that the size of the edit distance neighborhood is too large even in the best case:

$$|\mathcal{N}_r(\boldsymbol{x})| \geq \sum_{i=0}^{r} 255^i \sum_{j=i-r}^{r} \binom{|\boldsymbol{x}| + j}{i} \geq 255^r.$$

For example, applying the loosest bound that is independent of $\boldsymbol{x}$, we see that brute-force certification at radius $r = 128$ would require in excess of $255^r \approx 10^{308}$ queries to $f$. In contrast, our probabilistic certification mechanism (Figure 1) makes $n_{\mathrm{pred}} + n_{\mathrm{bnd}}$ queries to $f$, and we can provide high probability guarantees when the number of queries is of order $10^3$ or $10^4$.

## B  Proofs for Section 4

In this appendix, we provide proofs of the theoretical results stated in Section 4.

### B.1  Proof of Proposition 3

A sufficient condition for (8) is

$$\min_{\bar{\boldsymbol{x}} \in \mathcal{N}_r(\boldsymbol{x})} \min_{h \in \mathcal{F}(\boldsymbol{x})} p_y(\bar{\boldsymbol{x}}; h) \geq \max_{\bar{\boldsymbol{x}} \in \mathcal{N}_r(\boldsymbol{x})} \max_{h \in \mathcal{F}(\boldsymbol{x})} \left( \eta_y + \max_{y' \neq y} p_{y'}(\bar{\boldsymbol{x}}; h) - \min_{y' \neq y} \eta_{y'} \right). \qquad (17)$$

We first consider the multi-class case where $|\mathcal{Y}| > 2$. If $\eta_y \geq \min_{y' \neq y} \eta_{y'}$, then $p_y(\bar{\boldsymbol{x}}; h) \geq \max_{y' \neq y} p_{y'}(\bar{\boldsymbol{x}}; h)$ by (17) and we can upper-bound $\max_{y' \neq y} p_{y'}(\bar{\boldsymbol{x}}; h)$ by $\frac{1}{2}$. On the other hand, if $\eta_y \geq \min_{y' \neq y} \eta_{y'}$, we can only upper-bound $\max_{y' \neq y} p_{y'}(\bar{\boldsymbol{x}}; h)$ by 1. Thus when $|\mathcal{Y}| > 2$ (17) implies

$$\min_{\bar{\boldsymbol{x}} \in \mathcal{N}_r(\boldsymbol{x})} \min_{h \in \mathcal{F}(\boldsymbol{x})} p_y(\bar{\boldsymbol{x}}; h) \geq \begin{cases} \frac{1}{2} + \eta_y - \min_{y' \neq y} \eta_{y'}, & \eta_y \geq \min_{y' \neq y} \eta_{y'}, \\ 1 + \eta_y - \min_{y' \neq y} \eta_{y'}, & \eta_y < \min_{y' \neq y} \eta_{y'}. \end{cases}$$

Next, we consider the binary case where $|\mathcal{Y}| = 2$. Since the confidences sum to 1, we have $\max_{y' \neq y} p_{y'}(\bar{\boldsymbol{x}}; h) = 1 - p_y(\bar{\boldsymbol{x}}; h)$. Putting this in (17) implies

$$\min_{\bar{\boldsymbol{x}} \in \mathcal{N}_r(\boldsymbol{x})} \min_{h \in \mathcal{F}(\boldsymbol{x})} p_y(\bar{\boldsymbol{x}}; h) \geq \frac{1 + \eta_y - \min_{y' \neq y} \eta_{y'}}{2}.$$

### B.2  Proof of Lemma 4

Let $r_S \colon S \to \{1, \ldots, |S|\}$ be a bijection that returns the *rank* of an element in an ordered set $S$. Let $\dot{r}_S \colon 2^S \to 2^{\{1, \ldots, |S|\}}$ be an elementwise extension of $r_S$ that returns a *set of ranks* for an ordered set of elements—i.e., $\dot{r}_S(U) = \{ r_S(i) : i \in U \}$ for $U \subseteq S$. We claim $m(\bar{\epsilon}) = \dot{r}_{\epsilon^\star}^{-1}(\dot{r}_{\bar{\epsilon}^\star}(\bar{\epsilon}))$ is a bijection that satisfies the required property.

To prove the claim, we note that $m$ is a bijection from $2^{\bar{\epsilon}^\star}$ to $2^{\epsilon^\star}$ since it is a composition of bijections $\dot{r}_{\bar{\epsilon}^\star} \colon 2^{\bar{\epsilon}^\star} \to 2^{\{1, \ldots, l\}}$ and $\dot{r}_{\epsilon^\star}^{-1} \colon 2^{\{1, \ldots, l\}} \to 2^{\epsilon^\star}$ where $l = |\bar{\epsilon}^\star| = |\epsilon^\star|$. Next, we observe that $\dot{r}_{\bar{\epsilon}^\star}(\bar{\epsilon})$ relabels indices in $\bar{\epsilon}$ so they have the same effect when applied to $\boldsymbol{z}^\star$ as $\bar{\epsilon}$ on $\bar{\boldsymbol{x}}$ (this also holds for $\dot{r}_{\epsilon^\star}$ and $\epsilon$). Thus

$$\begin{aligned} \mathrm{apply}(\bar{\boldsymbol{x}}, \bar{\epsilon}) &= \mathrm{apply}(\boldsymbol{z}^\star, \dot{r}_{\bar{\epsilon}^\star}(\bar{\epsilon})) \\ &= \mathrm{apply}(\boldsymbol{z}^\star, \dot{r}_{\epsilon^\star}(\dot{r}_{\epsilon}^{-1}(\dot{r}_{\bar{\epsilon}^\star}(\bar{\epsilon})))) \\ &= \mathrm{apply}(\boldsymbol{x}, m(\bar{\epsilon})) \end{aligned}$$

as required. To prove the final statement, we use (4), (5) and (12) to write

$$
\frac{s(\bar{\epsilon}, \bar{x}; h)}{s(\epsilon, x; h)} = \frac{\mathbf{1}_{h(\mathrm{apply}(\bar{x}, \bar{\epsilon}))=y} p_{\mathrm{del}}^{|\bar{x}|-|\bar{\epsilon}|}(1-p_{\mathrm{del}})^{|\bar{\epsilon}|}}{\mathbf{1}_{h(\mathrm{apply}(x, \epsilon))=y} p_{\mathrm{del}}^{|x|-|\epsilon|}(1-p_{\mathrm{del}})^{|\epsilon|}}
$$

$$
= \frac{p_{\mathrm{del}}^{|\bar{x}|-|z|}(1-p_{\mathrm{del}})^{|z|} \mathbf{1}_{h(z)=y}}{p_{\mathrm{del}}^{|x|-|z|}(1-p_{\mathrm{del}})^{|z|} \mathbf{1}_{h(z)=y}}
$$

$$
= p_{\mathrm{del}}^{|\bar{x}|-|x|},
$$

where the second last line follows from the fact that $\mathrm{apply}(\bar{x}, \bar{\epsilon}) = \mathrm{apply}(x, \epsilon) = z$.

### B.3 Proof of Theorem 5

Let $\bar{\epsilon}^{\star}$ and $\epsilon^{\star}$ be defined as in Lemma 4. We derive an upper bound on the sum over $\epsilon \in 2^{\epsilon^{\star}}$ that appears in (13). Observe that

$$
\sum_{\epsilon \notin 2^{\epsilon^{\star}}} s(\epsilon, x; h) \leq \sum_{\epsilon \notin 2^{\epsilon^{\star}}} \Pr\left[G(x) = \epsilon\right]
$$

$$
= 1 - \sum_{\epsilon \in 2^{\epsilon^{\star}}} \Pr\left[G(x) = \epsilon\right]
$$

$$
= 1 - p_{\mathrm{del}}^{|x|-|\epsilon^{\star}|} \sum_{|\epsilon|=0}^{|\epsilon^{\star}|} \binom{|\epsilon^{\star}|}{|\epsilon|} p_{\mathrm{del}}^{|\epsilon^{\star}|-|\epsilon|}(1-p_{\mathrm{del}})^{|\epsilon|}
$$

$$
= 1 - p_{\mathrm{del}}^{|x|-|\epsilon^{\star}|}, \tag{18}
$$

where the first line follows from the inequality $\mathbf{1}_{h(\mathrm{apply}(x, \epsilon)=y)} \leq 1$; the second line follows from the law of total probability; the third line follows by constraining the indices $\{1, \ldots, |x|\} \setminus \bar{\epsilon}^{\star}$ to be deleted; and the last line follows from the normalization of the binomial distribution. Putting (18) and $\sum_{\bar{\epsilon} \in 2^{\bar{\epsilon}^{\star}}} s(\bar{\epsilon}, \bar{x}; h) \geq 0$ in (13) gives

$$
p_y(\bar{x}; h) \geq p_{\mathrm{del}}^{|\bar{x}|-|x|}\left(\mu_y - 1 - p_{\mathrm{del}}^{|x|-|\epsilon^{\star}|}\right)
$$

$$
= p_{\mathrm{del}}^{|\bar{x}|-|x|}\left(\mu_y - 1 - p_{\mathrm{del}}^{\frac{1}{2}(\mathrm{dist}_{\mathrm{LCS}}(\bar{x}, x)+|x|-|\bar{x}|)}\right). \tag{19}
$$

In the second line above we use the following relationship between the LCS distance and the length of the LCS $|z^{\star}| = |\epsilon^{\star}|$:

$$
\mathrm{dist}_{\mathrm{LCS}}(\bar{x}, x) = |\bar{x}| + |x| - 2|z^{\star}|.
$$

Since (19) is independent of the base classifier $h$, the lower bound on $\rho(\bar{x}, x, \mu_y)$ follows immediately.

### B.4 Proof of Corollary 6

Since the length of $x$ can only be changed by inserting or deleting elements in $\bar{x}$, we have

$$
|x| - |\bar{x}| = n_{\mathrm{ins}} - n_{\mathrm{del}}. \tag{20}
$$

We also observe that the LCS distance can be uniquely decomposed in terms of the counts of insertion ops $m_{\mathrm{ins}}$ and deletion ops $m_{\mathrm{del}}$: $\mathrm{dist}_{\mathrm{LCS}}(\bar{x}, x) = m_{\mathrm{del}} + m_{\mathrm{ins}}$. These counts can in turn be related to the given decomposition of edit ops counts for generalized edit distance. In particular, any substitution must be expressed as an insertion and deletion under LCS distance, which implies $m_{\mathrm{ins}} = n_{\mathrm{ins}} + n_{\mathrm{sub}}$ and $m_{\mathrm{del}} = n_{\mathrm{del}} + n_{\mathrm{sub}}$. Thus we have

$$
\mathrm{dist}_{\mathrm{LCS}}(\bar{x}, x) = n_{\mathrm{del}} + n_{\mathrm{ins}} + 2n_{\mathrm{sub}}. \tag{21}
$$

Substituting (20) and (21) in (14) gives the required result.

## B.5 Proof of Theorem 7

Eliminating $n_{\mathsf{sub}}$ from (16) using the constraint $n_{\mathsf{sub}} = r - n_{\mathsf{del}} - n_{\mathsf{ins}}$, we obtain a minimization problem in two variables:

$$\min_{n_{\mathsf{ins}}, n_{\mathsf{del}} \in \mathbb{N}_0} \quad \psi(n_{\mathsf{ins}}, n_{\mathsf{del}})$$

$$\text{s.t.} \qquad 0 \leq n_{\mathsf{ins}} + n_{\mathsf{del}} \leq r$$

where $\psi(n_{\mathsf{ins}}, n_{\mathsf{del}}) = p_{\mathsf{del}}^{n_{\mathsf{del}} - n_{\mathsf{ins}}} \left( \mu_y - 1 + p_{\mathsf{del}}^{r - n_{\mathsf{del}}} \right)$. Observe that $\psi$ is monotonically increasing in $n_{\mathsf{ins}}$ and $n_{\mathsf{del}}$:

$$\frac{\psi(n_{\mathsf{ins}} + 1, n_{\mathsf{del}})}{\psi(n_{\mathsf{ins}}, n_{\mathsf{del}})} = \frac{1}{p_{\mathsf{del}}} \geq 1$$

$$\frac{\psi(n_{\mathsf{ins}}, n_{\mathsf{del}} + 1)}{\psi(n_{\mathsf{ins}}, n_{\mathsf{del}})} = \frac{(\mu_y - 1)p_{\mathsf{del}}^{n_{\mathsf{del}} + 1} + p_{\mathsf{del}}^r}{(\mu_y - 1)p_{\mathsf{del}}^{n_{\mathsf{del}}} + p_{\mathsf{del}}^r} \geq 1,$$

where the second inequality follows since we only consider $r$ and $\mu_y$ such that the numerator and denominator are positive. Thus the minimizer is $(n_{\mathsf{ins}}^\star, n_{\mathsf{del}}^\star, n_{\mathsf{sub}}^\star) = (0, 0, r)$ and we find $\rho(\boldsymbol{x}; \mu_y) = \mu_y - 1 + p_{\mathsf{del}}^r$. The expression for the certified radius follows by solving $\rho(\boldsymbol{x}; \mu_y) \geq \nu_y(\boldsymbol{\eta})$ for non-negative integer $r$.

## B.6 Proof of Corollary 8

Recall that Corollary 6 gives the following lower bound on the classifier's confidence at $\boldsymbol{x}$:

$$\tilde{\rho}(\bar{\boldsymbol{x}}, \boldsymbol{x}, \mu_y) = p_{\mathsf{del}}^{n_{\mathsf{del}} - n_{\mathsf{ins}}} \left( \mu_y - 1 + p_{\mathsf{del}}^{n_{\mathsf{sub}} + n_{\mathsf{ins}}} \right).$$

Observe that we can replace $\mu_y$ by a lower bound $\underline{\mu}_y$ that holds with probability $1 - \alpha$ (as is done in lines 4–6 of Figure 1) and obtain a looser lower bound $\tilde{\rho}(\bar{\boldsymbol{x}}, \boldsymbol{x}, \underline{\mu}_y) \leq \tilde{\rho}(\bar{\boldsymbol{x}}, \boldsymbol{x}, \mu_y)$ that holds with probability $1 - \alpha$. Crucially, this looser lower bound has the same functional form, so all results depending on Corollary 6, namely Theorem 7 and Table 1, continue to hold albeit with probability $1 - \alpha$.

# C Background for malware detection case study

In this appendix, we provide background for our case study on malware detection, including motivation for studying certified robustness of malware detectors, a formulation of malware detection as a sequence classification problem, and a threat model for adversarial examples.

## C.1 Motivation

Malware (malicious software) detection is a vital capability for proactively defending against cyberattacks. Despite decades of progress, building and maintaining effective malware detection systems remains a challenge, as malware authors continually evolve their tactics to bypass detection and exploit new vulnerabilities. One technology that has lead to advancements in malware detection, is the application of machine learning (ML), which is now used in many commercial systems [69, 44, 46, 47] and continues to be an area of interest in the malware research community [70, 71, 45, 50]. While traditional detection techniques rely on manually-curated signatures or detection rules, ML allows a detection model to be learned from a training corpus, that can potentially generalize to unseen programs.

Although ML has an apparent advantage in detecting previously unseen malware, recent research has shown that ML-based static malware detectors can be evaded by applying adversarial perturbations [18–20, 37, 51, 31, 52, 17, 39]. A variety of perturbations have been considered with different effects at the semantic level, however all of them can be modeled as inserting, deleting and/or substituting bytes. This prompts us to advance certified robustness for sequence classifiers within this general threat model—where an attacker can perform byte-level edits.

## C.2 Related work

Several empirical defense methods have been proposed to improve robustness of ML classifiers [63, 64]. Incer Romeo et al. [34] compose manually crafted Boolean features with a classifier that is constrained to be monotonically increasing with respect to selected inputs. This approach permits a combination of (potentially vulnerable) learned behavior with domain knowledge, and thereby aims to mitigate adversarial examples. Demontis et al. [63] show that the sensitivity of linear support vector machines to adversarial perturbations can be reduced by training with $\ell_\infty$ regularization of weights. In another work, Quiring et al. [64] take advantage of heuristic-based semantic gap detectors and an ensemble of feature classifiers to improve empirical robustness. Compared to our work on certified adversarial defenses, these approaches do not provide formal guarantees.

*Binary normalization* [72–75] was originally proposed to defend against polymorphic/metamorphic malware, and can also be seen as a mitigation to certain adversarial examples. It attempts to sanitize binary obfuscation techniques by mapping malware to a canonical form before running a detection algorithm. However, binary normalization cannot fully mitigate attacks like *Disp* (see Table 9), as deducing opaque and evasive predicates are NP-hard problems [17].

Dynamic analysis can provide additional insights for malware detection. In particular, it can record a program's behavior while executing it in a sandbox (e.g., collecting a call graph or network traffic) [76–80]. Though detectors built on top of dynamic analysis can be more difficult to evade, as the attacker needs to obfuscate the program's behavior, they are still susceptible to adversarial perturbations. For example, an attacker may insert API calls to obfuscate a malware's behavior [81–84]. Applying RS-Del to certify detectors that operate on call sequences [80] or more general dynamic features would be an interesting future direction.

## C.3 Static ML-based malware detection

We formulate malware detection as a sequence classification problem, where the objective is to classify a file in its raw byte sequence representation as malicious or benign. In the notation of Section 2, we assume the space of input sequences (files) is $\mathcal{X} = \Omega^\star$ where $\Omega = \{0, 1, \ldots, 255\}$ denotes the set of bytes, and we assume the set of classes is $\mathcal{Y} = \{0, 1\}$ where 1 denotes the 'malicious' class and 0 denotes the 'benign' class. Within this context, a *malware detector* is simply a classifier $f : \mathcal{X} \to \mathcal{Y}$.

**Detector assumptions** Malware detectors are often categorized according to whether they perform static or dynamic analysis. Static analysis extracts information without executing code, whereas dynamic analysis extracts information by executing code and monitoring its behavior. In this work, we focus on machine learning-based static malware detectors, where the ability to extract and synthesize information is learned from data. Such detectors are suitable as base classifiers for RS-Del, as they can learn to make (weak) predictions for incomplete files where chunks of bytes are arbitrarily removed. We note that dynamic malware detectors are not compatible with RS-Del, since it is not generally possible to execute an incomplete file.

**Incorporating semantics** In Section 3.3, we noted that our methods are compatible with *sequence chunking* where the original input sequence is partitioned into chunks, and reinterpreted as a sequence of chunks rather than a sequence of lower-level elements. In the context of malware detection, we can partition a byte sequence into semantically meaningful chunks using information from a disassembler, such as Ghidra [85]. For example, a disassembler can be applied to a Windows executable to identify chunks of raw bytes that correspond to components of the header, machine instructions, raw data, padding etc. Applying our deletion smoothing mechanism at the level of semantic chunks, rather than raw bytes, may improve robustness as it excludes edits within chunks that may be semantically invalid. It also yields a different chunk-level edit distance certificate, that may cover a larger set of adversarial examples than a byte-level certificate of the same radius. Figure 3 illustrates the difference between byte-level and chunk-level deletion for a Windows executable, where chunks correspond to machine instructions (such as `push ebp`) or non-instructions (`NI`).

**Original file**

| File offset | Byte | Instruction chunks | |
|---|---|---|---|
| 00000000 | 77 | NI | |
| 00000001 | 90 | NI | |
| 00000002 | 144 | NI | |
| ⋮ | ⋮ | ⋮ | |
| 00000400 | 85 | push | ebp |
| 00000401 | 139 | mov | ebp, esp |
| 00000402 | 236 | | |
| 00000403 | 131 | | |
| 00000404 | 236 | sub | esp, 5Ch |
| 00000405 | 92 | | |
| ⋮ | ⋮ | ⋮ | |

**File under byte-level deletion (BYTE)**

| File offset | Byte |
|---|---|
| 00000000 | 77 |
| 00000002 | 144 |
| ⋮ | ⋮ |
| 00000400 | 85 |
| 00000403 | 131 |
| 00000404 | 236 |
| ⋮ | ⋮ |

**File under chunk-level deletion (INSN)**

| File offset | Chunk |
|---|---|
| 00000001 | 90 |
| ⋮ | ⋮ |
| 00000400 | 85 |
| 00000401 | 139 236 |
| ⋮ | |

Figure 3: Illustration of the deletion smoothing mechanism applied to an executable file at the byte-level versus chunk-level. *Left:* An executable file where the elementary byte sequence representation is shown in the 2nd column and chunks that correspond to machine instructions are shown in the 3rd column (sourced from the Ghidra [85] disassembler). Bytes that do not correspond to machine instructions are marked NI. Shading represents bytes (light gray) or instruction chunks (dark gray) that are deleted in the corresponding perturbed file to the right. *Middle:* A perturbed file produced by the deletion mechanism operating at the byte level (BYTE). Notice that individual instructions may be partially deleted. *Right:* A perturbed file produced by the deletion mechanism operating at the chunk-level (INSN).

## C.4   Threat model

We next specify the modeled attacker's goals, capabilities and knowledge for our malware detection case study [86].

**Attacker's objective**   We consider evasion attacks against a malware detector $f : \mathcal{X} \to \mathcal{Y}$, where the attacker's objective is to transform an executable file $x$ so that it is misclassified by $f$. To ensure the attacked file $\bar{x}$ is useful after evading detection, we require that it is *functionally equivalent* to the original file $x$. We focus on evasion attacks that aim to misclassify a *malicious* file as *benign* in our experiments, as these attacks dominate prior work [52]. However, the robustness certificates derived in Section 4 also cover attacks in the opposite direction—where a *benign* file is misclassified as *malicious*.

**Attacker's capability**   We measure the attacker's capability in terms of the number of elementary edits they make to the original file $x$. If the attacker is capable of making up to $c$ elementary edits, then they can transform $x$ into any file in the edit distance ball of radius $c$ centred on $x$:

$$\mathcal{A}_c(x) = \{\bar{x} \in \mathcal{X} : \mathrm{dist}_O(x, \bar{x}) \leq c\}.$$

Here $\mathrm{dist}_O(x, \bar{x})$ denotes the edit distance from the original file $x$ to the attacked file $\bar{x}$ under the set of edit operations (ops) $O$. We assume $O$ consists of elementary byte-level or chunk-level deletions (del), insertions (ins) and substitutions (sub), or a subset of these operations.

We note that edit distance is a reasonable proxy for the cost of running evasion attacks that iteratively apply localized functionality-preserving edits (e.g., [19, 37, 38, 17, 39]). For these attacks, the edit distance scales roughly linearly with the number of attack iterations, and therefore the attacker has an incentive to minimize edit distance. While attacks do exist that make millions of elementary edits in the malware domain (e.g., [31]), we believe that an edit distance-constrained threat model is an important step towards realistic threat models for certified malware detection. (To examine the effect of large edits on robustness we include the *GAMMA* attack [31] in experiments covered in Appendix F.)

*Remark* 9. The set $\mathcal{A}_c(x)$ *overestimates* the capability of an edit distance-constrained attacker, because it may include files that are not functionally equivalent to $x$. For example, $\mathcal{A}_c(x)$ may include files that are not malicious (assuming $x$ is malicious) or files that are invalid executables. This poses no problem for certification, since overestimating an attacker's capability merely leads to a stronger certificate than required. Indeed, overestimating the attacker's capability seem necessary, as functionally equivalent files are difficult to specify, let alone analyze.

Table 3: Summary of datasets.

| Dataset | Label | Number of samples | | |
| --- | --- | --- | --- | --- |
| | | Train | Validation | Test |
| Sleipnir2 | Benign | 20 948 | 7 012 | 6 999 |
| | Malicious | 20 768 | 6 892 | 6 905 |
| VTFeed | Benign | 111 258 | 13 961 | 13 926 |
| | Malicious | 111 395 | 13 870 | 13 906 |

**Attacker's knowledge** In our certification experiments in Appendix E, we assume the attacker has full knowledge of the malware detector and certification scheme. When testing published attacks in Appendix F, we consider both white-box and black-box access to the malware detector. In the black-box setting, the attacker may make an unlimited number of queries to the malware detector without observing its internal operation. We permit access to detection confidence scores, which are returned alongside predictions even in the black-box setting. In the white-box setting, the attacker can additionally inspect the malware detector's source code. Such a strong assumption is needed for white-box attacks against neural network-based detectors that compute loss gradients with respect to the network's internal representation of the input file [20, 17].

## D   Experimental setup for malware detection case study

In this appendix, we detail the experimental setup for our malware detection case study.

### D.1   Datasets

Though our methods are compatible with executable files of any format, in our experiments we focus on the *Portable Executable (PE) format* [87], since datasets, malware detection models and adversarial attacks are more extensively available for this format. Moreover, PE format is the standard for executables, object files and shared libraries in the Microsoft Windows operating system, making it an attractive target for malware authors. We use two PE datasets which are summarized in Table 3 and described below.

**Sleipnir2** This dataset attempts to replicate data used in past work [54], which was not published with raw samples. We were able to obtain the raw malicious samples from a public malware repository called VirusShare [88] using the provided hashes. However, since there is no similar public repository for benign samples, we followed established protocols [89, 90, 20] to collect a new set of benign samples. Specifically, we set up a Windows 7 virtual machine with over 300 packages installed using Chocolatey package manager [91]. We then extracted PE files from the virtual machine, which were assumed benign[4], and subsampled them to match the number of malicious samples. The dataset is randomly split into training, validation and test sets with a ratio of 60%, 20% and 20% respectively.

**VTFeed** This dataset was first used in recent attacks on end-to-end ML-based malware detectors [17]. It was collected from VirusTotal—a commercial threat intelligence service—by sampling PE files from the live feed over a period of two weeks in 2020. Labels for the files were derived from the 68 antivirus (AV) products aggregated on VirusTotal at the time of collection. Files were labeled *malicious* if they were flagged malicious by 40 or more of the AV products, they were labeled *benign* if they were not flagged malicious by any of the AV products, and any remaining files were excluded. Following Lucas et al. [17], the dataset is randomly split into training, validation and test sets with a ratio of 80%, 10%, and 10% respectively.

We note that VTFeed comes with strict terms of use, which prohibit us from loading it on our high performance computing (HPC) cluster. As a result, we use Sleipnir2 for comprehensive experiments (e.g., varying $p_{\text{del}}$, $\eta$) on the HPC cluster, and VTFeed for a smaller selection of experiments run on a local server.

---

[4]Chocolatey packages are validated against VirusTotal [92].

## D.2 Malware detection models

We experiment with malware detection models based on MalConv [48]. MalConv was one of the first *end-to-end* neural network models proposed for malware detection—i.e., it learns to classify directly from raw byte sequences, rather than relying on manually engineered features. Architecturally, it composes a learnable embedding layer with a shallow convolutional network. A large window size and stride of 500 bytes are employed to facilitate scaling to long byte sequences. Though MalConv is compatible with arbitrarily long byte sequences in principle, we truncate all inputs to 2MB to support training efficiency. We use the original parameter settings and training procedure [48], except where specified in Appendix D.5.

Using MalConv as a basis, we consider three models as described below.

**NS** A vanilla non-smoothed (NS) MalConv model. This model serves as a non-certified, non-robust baseline—i.e., no specific techniques are employed to improve robustness to evasion attacks and certification is not supported.

**RS-Abn** A smoothed MalConv model using the *randomized ablation* smoothing mechanism proposed by Levine and Feizi [24] and reviewed in Appendix I. This model serves as a certified robust baseline, albeit covering a more restricted threat model than the edit distance threat model we propose in Section 2. Specifically, it supports robustness certification for the Hamming distance threat model, where the adversary is limited to substitution edits ($O = \{\mathsf{sub}\}$). Since Levine and Feizi's formulation is for images, several modifications are required to support malware detection as described in Appendix G. To improve convergence, we also apply gradient clipping when learning parameters in the embedding layer (see Appendix G). We consider variants of this model for different values of the ablation probability $p_{\mathsf{ab}}$.

**RS-Del** A smoothed MalConv model using our proposed randomized deletion smoothing mechanism. This model supports robustness certification for the generalized edit distance threat model where $O \subseteq \{\mathsf{del}, \mathsf{ins}, \mathsf{sub}\}$. We consider variants of this model for different values of the deletion probability $p_{\mathsf{del}}$, decision thresholds $\eta$, and whether deletion/certification is performed at the byte-level (BYTE) or chunk-level (INSN). We perform chunking as illustrated in Figure 3—i.e., we chunk bytes that correspond to distinct machine instructions using the Ghidra disassembler.

## D.3 Controlling false positive rates

Malware detectors are typically tuned to achieve a low false positive rate (FPR) (e.g., less than 0.1–1%) since producing too many false alarms is a nuisance to users.[5] To make all malware detection models comparable, we calibrate the FPR to 0.5% on the test set for the experiments reported in Appendix E and 0.5% on the validation set for the experiments reported in Appendix F unless otherwise noted. This calibration is done by adjusting the decision threshold of the base MalConv model.

## D.4 Compute resources

Experiments for the Sleipnir2 dataset were run a high performance computing (HPC) cluster, where the requested resources varied depending on the experiment. We generally requested a single NVIDIA P100 GPU when training and certifying models. Experiments for the VTFeed dataset were run on a local server due to restrictive terms of use. Compute resources and approximate wall clock running times are reported in Tables 4 and 5 for training and certification for selected parameter settings. Running times for other parameters settings are lower than the ones reported in these tables.

## D.5 Parameter settings

We specify the parameter settings and training procedure for MalConv, which is used standalone in NS, and as a base model for the smoothed models RS-Del and RS-Abn. Table 6 summarizes our setup, which is consistent across all three models except where specified. We follow the authors of MalConv [48] when setting parameters for the model and the optimizer, however we set a larger

---

[5]`https://www.av-comparatives.org/testmethod/false-alarm-tests/`

Table 4: Compute resources used for training. Note that the wall clock times reported here are for an unoptimized implementation where the smoothing mechanism is executed on the CPU.

| Dataset | Requested resources | Model | Parameters | Time | Notes |
|---------|--------------------|-------|-----------|------|-------|
| Sleipnir2 | 1 NVIDIA P100 GPU, 4 cores on Intel Xeon Gold 6326 CPU | NS | – | 22 hr | Trained for 50 epochs, converged in 15 epochs |
| | | RS-Del | BYTE, $p_{\text{del}} = 95\%$ | 39 hr | Trained for 100 epochs, converged in 50 epochs |
| | | RS-Del | INSN, $p_{\text{del}} = 95\%$ | 48 hr | Trained for 100 epochs, converged in 20 epochs |
| | | RS-Abn | $p_{\text{ab}} = 95\%$ | 40 hr | Trained for 100 epochs, converged in 90 epochs |
| VTFeed | 1 NVIDIA RTX3090 GPU, 6 cores on AMD Ryzen Threadripper PRO 3975WX CPU | NS | – | 139 hr | Trained for 100 epochs, converged in 25 epochs |
| | | RS-Del | BYTE, $p_{\text{del}} = 97\%$ | 152 hr | Trained for 100 epochs, converged in 20 epochs |

Table 5: Compute resources used for certification on the test set. The evaluation dataset is partitioned and processed on multiple compute nodes with the same specifications. The reported time is the sum of wall times on each compute node. Note that the times reported are for an unoptimized implementation where the smoothing mechanism is executed on the CPU.

| Dataset | Requested resources | Model | Parameters | Time |
|---------|--------------------|-------|-----------|------|
| Sleipnir2 | 1 NVIDIA P100 GPU, 12 cores on Intel Xeon Gold 6326 CPU | NS | – | 5 min |
| | | RS-Del | BYTE, $p_{\text{del}} = 95\%$ | 65 hr |
| | | RS-Del | INSN, $p_{\text{del}} = 95\%$ | 140 hr |
| | | RS-Abn | $p_{\text{ab}} = 95\%$ | 210 hr |
| VTFeed | 1 NVIDIA RTX3090 GPU, 6 cores on AMD Ryzen Threadripper PRO 3975WX CPU | NS | – | 4 min |
| | | RS-Del | BYTE, $p_{\text{del}} = 97\%$ | 500 hr |

maximum input size of 2MiB to accommodate larger inputs without clipping. Due to differences in available GPU memory for the Sleipnir2 and VTFeed experiments, we use a larger batch size for VTFeed than for Sleipnir2. We also set a higher limit on the maximum number of epochs for VTFeed, as it is a larger dataset, although the NS and RS-Del models converge within 50 epochs for both datasets. To stabilize training for the smoothed models (RS-Del and RS-Abn), we modify the smoothing mechanisms during *training only* to ensure at least 500 raw bytes are preserved. This may limit the number of deletions for RS-Del and the number of ablated (masked) bytes for RS-Abn. For RS-Abn, we clip the gradients for the embedding layer to improve convergence (see Appendix G).

## E    Evaluation of robustness certificates for malware detection

In this appendix, we evaluate the robustness guarantees and accuracy of RS-Del for malware detection. We consider two instantiations of the edit distance threat model. First, in Appendix E.1, we consider the Levenshtein distance threat model, where the attacker's elementary edits are unconstrained and may include deletions, insertions and substitutions. Then, in Appendix E.2, we consider the more restricted Hamming distance threat model, where an attacker is only able to perform substitutions. We summarize our findings in Appendix E.3. Overall, we find that RS-Del generates robust predictions with minimal impact on model accuracy for the Levenshtein distance threat model, and outperforms RS-Abn [24] for the Hamming distance threat model.

Table 6: Parameter settings for MalConv, the optimizer and training procedure. Parameter settings are consistent across all malware detection models (NS, RS-Del, RS-Abn) except where specified.

| | Parameter | Values |
|---|---|---|
| MalConv | Max input size | 2097152 |
| | Embedding size | 8 |
| | Window size | 500 |
| | Channels | 128 |
| Optimizer | Python class | `torch.optim.SGD` |
| | Learning rate | 0.01 |
| | Momentum | 0.9 |
| | Weight decay | 0.001 |
| Training | Batch size | 24 (Sleipnir2), 32 (VTFeed) |
| | Max. epoch | 50 (Sleipnir2), 100 (VTFeed) |
| | Min. preserved bytes | 500 (RS-Del, RS-Abn), NA (NS) |
| | Embedding gradient clipping | 0.5 (RS-Abn), $\infty$ (RS-Del, NS) |
| | Early stopping | If validation loss does not improve after 10 epochs |

We report the following quantities in our evaluation:

- *Certified radius (CR).* The radius of the largest robustness certificate that can be issued for a given input, model and certification method. Note that this is a conservative measure of robustness since it is *tied to the certification method*. The *median CR* is reported on the test set.

- *Certified accuracy* [13, 14], also known as *verified-robust accuracy* [66, 55], evaluates robustness certificates and accuracy of a model simultaneously with respect to a test set. It is defined as the fraction of instances in the test set $\mathbb{D}$ for which the model $f$'s prediction is correct *and* certified robust at radius $r$ or greater:

$$\text{CERTACC}_r(\mathbb{D}) = \sum_{(\boldsymbol{x}, y) \in \mathbb{D}} \frac{\mathbf{1}_{f(\boldsymbol{x})=y} \mathbf{1}_{\text{CR}(\boldsymbol{x}) \geq r}}{|\mathbb{D}|} \tag{22}$$

  where $\text{CR}(\boldsymbol{x})$ denotes the certified radius for input $\boldsymbol{x}$ returned by the certification method.

- *Clean accuracy.* The fraction of instances in the test set for which the model's prediction is correct.

We briefly mention default parameter settings for the experiments presented in this appendix. When approximating the smoothed models (RS-Del and RS-Abn) we sample $n_{\text{pred}} = 1000$ perturbed inputs for prediction and $n_{\text{bnd}} = 4000$ perturbed inputs for certification, while setting the significance level $\alpha$ to 0.05. Unless otherwise specified, we set the decision thresholds for the smoothed models so that $\boldsymbol{\eta} = 0$. After fixing $\boldsymbol{\eta}$, the decision thresholds for the base models are tuned to yield a false positive rate of 0.5%. We note that the entire test set is used when reporting metrics and summary statistics in this appendix.

### E.1 Levenshtein distance threat model

We first present results for the Levenshtein distance threat model, where the attacker's elementary edits are unconstrained ($O = \{\text{del}, \text{ins}, \text{sub}\}$). We vary three parameters associated with RS-Del: the deletion probability $p_{\text{del}}$, the decision thresholds of the smoothed model $\boldsymbol{\eta}$, and the level of sequence chunking (i.e., whether sequences are chunked at the byte-level or instruction-level). We use NS as a baseline as there are no prior certified defenses for the Levenshtein distance threat model to our knowledge.

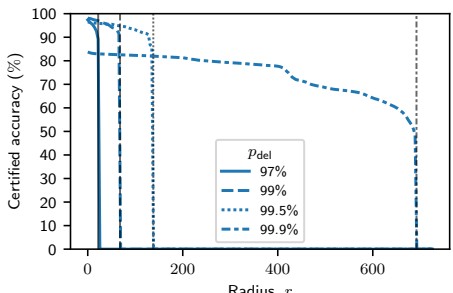 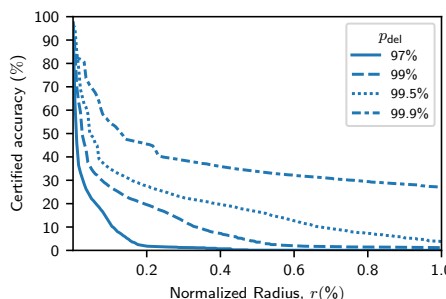

Figure 4: Certified accuracy for RS-Del as a function of the radius in bytes (left horizontal axis), radius normalized by file size (right horizontal axis) and byte deletion probability $p_{\text{del}}$ (line styles). The results are plotted for the Sleipnir2 test set under the byte-level Levenshtein distance threat model (with $O = \{\text{del}, \text{ins}, \text{sub}\}$). The grey vertical lines in the left plot represent the best achievable certified radius for RS-Del (setting $\mu_y = 1$ in the expressions in Table 1).

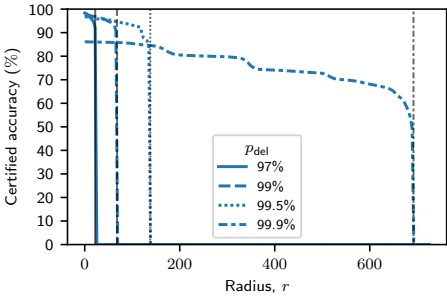 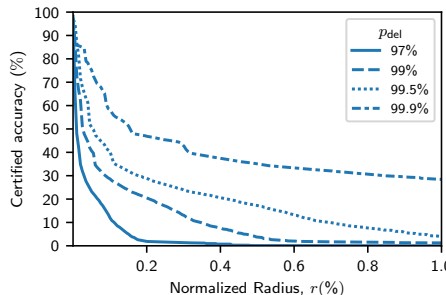

Figure 5: Certified accuracy for RS-Del with chunk-level deletion (INSN) as a function of the radius in chunks (left horizontal axis), radius normalized by sequence length in chunks (right horizontal axis) and chunk deletion probability $p_{\text{del}}$ (line styles). The results are plotted for the Sleipnir2 test set under the chunk-level Levenshtein distance threat model (with $O = \{\text{del}, \text{ins}, \text{sub}\}$). The grey vertical lines in the left plot represent the best achievable certified radius for RS-Del (setting $\mu_y = 1$ in the expressions in Table 1).

**Certified accuracy** Figure 4 plots the certified accuracy of RS-Del using byte-level deletion as a function of the radius (left horizontal axis), radius normalized by file size (right horizontal axis) on the Sleipnir2 dataset for several values of $p_{\text{del}}$. We observe that the curves for larger values of $p_{\text{del}}$ approximately dominate the curves for smaller values of $p_{\text{del}}$, for $p_{\text{del}} \leq 99.5\%$ (i.e., the accuracy is higher or close for all radii). This suggests that the robustness of RS-Del can be improved without sacrificing accuracy by increasing $p_{\text{del}}$ up to 99.5%. However, for the larger value $p_{\text{del}} = 99.9\%$, we observe a drop in certified accuracy of around 10% for smaller radii and an increase for larger radii. By normalizing with respect to the file size, we can see that our certificate is able to certify up to 1% of the file size. We also include an analogous plot for chunk-level deletion in Figure 5 which demonstrates similar behavior. We note that chunk-level deletion arguably provides stronger guarantees, since the effective radius for chunk-level Levenshtein distance is larger than for byte-level Levenshtein distance.

It is interesting to relate these certification results to published evasion attacks. Figure 4 shows that we can achieve a certified accuracy in excess of 90% at a Levenshtein distance radius of 128 bytes when $p_{\text{del}} = 99.5\%$. This radius is larger than the median Levenshtein distance of two attacks that manipulate headers of PE files [37, 38] (see Table 9 in Appendix F). We can therefore provide reasonable robustness guarantees against these two attacks. However, a radius of 128 bytes is orders of magnitude smaller than the median Levenshtein distances of other published attacks which range from tens of KB [17, 20] to several MB [31] (also reported in Table 9). While some of these attacks

Table 7: Clean accuracy and robustness metrics for RS-Del as a function of the dataset (Sleipnir2 and VTFeed), deletion probability $p_{\text{del}}$ and deletion level (BYTE or INSN). All metrics are computed on the test set. Here "abstain rate" refers to the fraction of test instances for which RS-Del abstains (line 7 in Figure 1), and "UB" refers to an upper bound on the median CR for a best case smoothed model (based on Table 1 with $\mu_y = 1$). A good tradeoff is achieved when $p_{\text{del}} = 99.5\%$ for both the byte-level (BYTE) and chunk-level (INSN) certificates (highlighted in bold face below).

| Dataset | Model | Parameters | Clean accuracy (Abstain rate) % | Median CR (UB) | | Median NCR % |
|---|---|---|---|---|---|---|
| Sleipnir2 | NS | – | 98.9 – | – | – | – |
| | RS-Del | BYTE, $p_{\text{del}} = 90\%$ | 97.1 (0.2) | 6 | (6) | 0.0023 |
| | | BYTE, $p_{\text{del}} = 95\%$ | 97.8 (0.0) | 13 | (13) | 0.0052 |
| | | BYTE, $p_{\text{del}} = 97\%$ | 97.4 (0.1) | 22 | (22) | 0.0093 |
| | | BYTE, $p_{\text{del}} = 99\%$ | 98.1 (0.1) | 68 | (68) | 0.0262 |
| | | **BYTE, $p_{\text{del}} = 99.5\%$** | **96.5 (0.2)** | **137** | **(138)** | **0.0555** |
| | | BYTE, $p_{\text{del}} = 99.9\%$ | 83.7 (3.4) | 688 | (692) | 0.2269 |
| | | INSN, $p_{\text{del}} = 90\%$ | 97.9 (0.1) | 6 | (6) | 0.0026 |
| | | INSN, $p_{\text{del}} = 95\%$ | 97.8 (0.1) | 13 | (13) | 0.0056 |
| | | INSN, $p_{\text{del}} = 97\%$ | 98.3 (0.0) | 22 | (22) | 0.0095 |
| | | INSN, $p_{\text{del}} = 99\%$ | 97.6 (0.1) | 68 | (68) | 0.0292 |
| | | **INSN, $p_{\text{del}} = 99.5\%$** | **96.8 (0.2)** | **137** | **(138)** | **0.0589** |
| | | INSN, $p_{\text{del}} = 99.9\%$ | 86.1 (0.2) | 689 | (692) | 0.2982 |
| VTFeed | NS | – | 98.9 – | – | – | – |
| | RS-Del | BYTE, $p_{\text{del}} = 97\%$ | 92.1 (0.9) | 22 | (22) | 0.0045 |
| | | BYTE, $p_{\text{del}} = 99\%$ | 86.9 (0.8) | 68 | (68) | 0.0122 |

arguably fall outside an edit distance constrained threat model, we consider them in our empirical evaluation of robustness in Appendix F.

**Clean accuracy and abstention rates**  Table 7 reports clean accuracy for RS-Del and the non-certified NS baseline. It also reports abstention rates for RS-Del, the median certified radius (CR), and the median certified radius normalized by file size (NCR). We find that clean accuracy for Sleipnir2 follows similar trends as certified accuracy: it is relatively stable for $p_{\text{del}}$ in the range 90–99.5%, but drops by more than 10% at $p_{\text{del}} = 99.9\%$. We note that the clean accuracy of RS-Del (excluding $p_{\text{del}} = 99.9\%$) is at most 3% lower than the NS baseline for Sleipnir2 and at most 7% lower than the NS baseline for VTFeed. We observe minimal differences in the results for chunk-level (INSN) and byte-level (BYTE) deletion smoothing, but note that the effective CR is larger for chunk-level smoothing, since each chunk may contain several bytes.

**Accuracy under high deletion**  It may be surprising that RS-Del can maintain high accuracy even when deletion is aggressive. We offer some possible explanations. First, we note that even with a high deletion probability of $p_{\text{del}} = 99.9\%$, the smoothed model accesses almost all of the file in expectation, as it aggregates $n_{\text{pred}} = 1000$ predictions from the base model each of which accesses a random 0.1% of the file in expectation. Second, we posit that malware detection may be "easy" for RS-Del on these datasets. This could be due to the presence of signals that are robust to deletion (e.g., file size or byte frequencies) or redundancy of signals (i.e., if a signal is deleted in one place it may be seen elsewhere).

**Decision threshold**  We demonstrate how the decision thresholds $\boldsymbol{\eta}$ introduced in Section 3.1 can be used to trade off certification guarantees between classes. We consider normalized decision thresholds where $\sum_y \eta_y = 1$ and $\eta_y \in [0, 1]$. We only specify the value of $\eta_1$ when discussing our results, noting that $\eta_0 = 1 - \eta_1$ in our two-class setting.

Table 8 provides error rates and robustness metrics for several values of $\eta_1$, using byte-level Levenshtein distance with $p_{\text{del}} = 99.5\%$. When varying $\eta_1$, we also vary the decision threshold of the base model to achieve a target false positive rate (FPR) of 0.5%. Looking at the table, we see that $\eta_1$ has minimal impact on the false negative rate (FNR), which is stable around 7%. However, there

Table 8: Impact of the smoothed decision threshold $\eta_1$ on false negative error rate (FNR) and median certified radius (CR) for malicious and benign files. The false positive rate (FPR) is set to a target value of 0.5% by varying the decision threshold of the base model. The results are reported for Sleipnir2 with $p_{del} = 99.5\%$ using byte-level Levenshtein distance. "UB" refers to an upper bound on the median CR for a best case smoothed model (based on Table 1 with $\mu_y = 1$).

| | | | Median CR (UB) | | | |
|---|---|---|---|---|---|---|
| $\eta_1$ (%) | FNR (%) | FPR (%) | Malicious | | Benign | |
| 50 | 6.8 | 0.5 | 137 | (138) | 137 | (138) |
| 25 | 6.9 | 0.5 | 275 | (276) | 57 | (57) |
| 10 | 6.8 | 0.5 | 455 | (459) | 20 | (21) |
| 5 | 6.6 | 0.5 | 578 | (597) | 10 | (10) |
| 1 | 7.1 | 0.5 | 582 | (918) | 1 | (2) |
| 0.5 | 6.9 | 0.5 | 506 | (1057) | 0 | (0) |

is a significant impact on the median CR (and theoretical upper bound), as reported separately for each class. The median CR is balanced for both the malicious and benign class when $\eta_1 = 50\%$, but favours the malicious class as $\eta_1$ is decreased. For instance when $\eta_1 = 5\%$ a significantly larger median CR is possible for malicious files (137 to 578) at the expense of the median CR for benign files (137 to 10). This asymmetry in the class-specific CR is a feature of the theory—that is, in addition to controlling a tradeoff between error rates of each class, $\eta_1$ also controls a tradeoff between the CR for each class (see Table 1).

Figure 6 plots the certified true positive rate (TPR) and true negative rate (TNR) of RS-Del on the Sleipnir2 dataset for several values of $\eta_1$. The certified TPR and TNR can be interpreted as class-specific analogues of the certified accuracy. Concretely, the certified TPR (TNR) at radius $r$ is the fraction of malicious (benign) instances in the test set for which the model's prediction is correct *and* certified robust at radius $r$. The certified TPR and TNR jointly measure accuracy and robustness and complement the metrics reported in Table 8. Looking at Figure 6, we see that the certified TNR curves drop more rapidly to zero than the certified TPR curves as $\eta_1$ decreases. Again, this suggests decreasing $\eta_1$ sacrifices the certified radii of benign instances to increase the certified radii of malicious instances. We note that the curves for $\eta_1 = 50\%$ correspond to the same setting as the certified accuracy curve in Figure 4 (with $p_{del} = 99.5\%$).

## E.2 Hamming distance threat model

We now turn to the more restricted Hamming distance threat model, where the attacker is limited to performing substitutions only ($O = \{\text{sub}\}$). We choose to evaluate this threat model as it is covered in previous work on randomized smoothing, called *randomized ablation* [24] (abbreviated RS-Abn), and can serve as a baseline for comparison with our method. Recall that we adapt RS-Abn for malware detection by introducing a parameter called $p_{ab}$, which is the fraction of bytes that are "ablated" (replaced by a special masked value) (see Appendix D.2). This parameter is analogous to $p_{del}$ in RS-Del, except that the number of ablated bytes is deterministic in RS-Abn, whereas the number of deleted bytes is random in RS-Del. We compare RS-Del and RS-Abn for varying values of $p_{del}$ and $p_{ab}$ using the Sleipnir2 dataset and byte-level Hamming distance.

**Certified accuracy**  Figure 2 plots the certified accuracy of RS-Del and RS-Abn for three values of $p_{del}$ and $p_{ab}$. We observe that the certified accuracy is uniformly larger for our proposed method RS-Del than for RS-Abn when $p_{del} = p_{ab}$. The superior certification performance of RS-Del is somewhat surprising given it is not optimized for the Hamming distance threat model. One possible explanation relates to the learning difficulty of RS-Abn compared with RS-Del. Specifically, we find that stochastic gradient descent is slower to converge for RS-Abn despite our attempts to improve convergence (see Appendix G). Recall, that RS-Del provides certificates for any of the threat models in Table 1—in addition to the Hamming distance certificate—without needing to modify the smoothing mechanism.

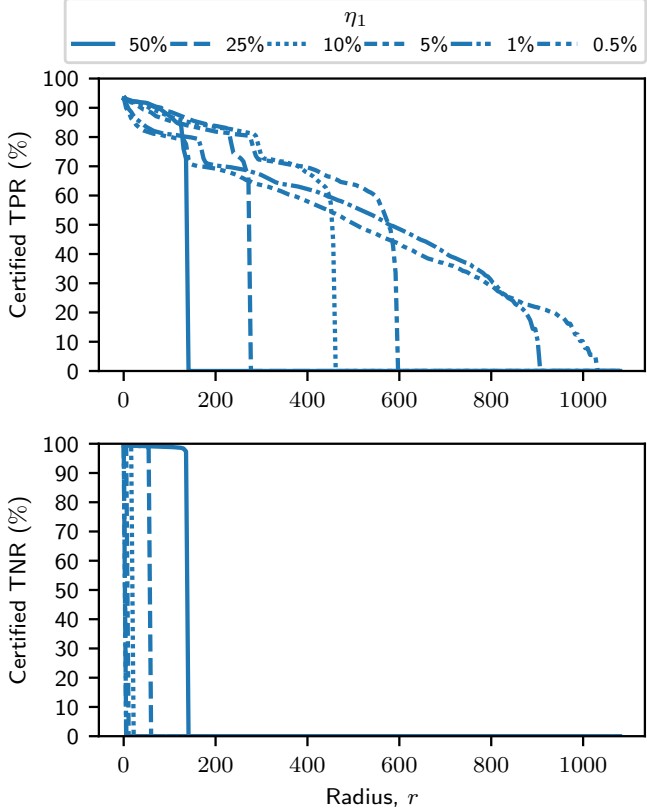

Figure 6: Certified true positive rate (TPR) and true negative rate (TNR) of RS-Del as a function of the certificate radius $r$ (horizontal axis) and the decision threshold $\eta_1$ (line style). The results are plotted for the Sleipnir2 test set for byte-level deletion (BYTE) with $p_{\text{del}} = 99.5\%$ under the Levenshtein distance threat model (with $O = \{\text{del}, \text{ins}, \text{sub}\}$). It is apparent that $\eta_1$ controls a tradeoff in the certified radius between the malicious (measured by TPR) and benign (measured by TNR) classes. Note that in this setting, a non-smoothed, non-certified model (NS) achieves a clean TPR and TNR of 98.2% and 99.5% respectively.

**Tightness**  RS-Abn is provably tight, in the sense that it is not possible to issue a larger Hamming distance certificate unless more information is made available to the certification mechanism or the ablation smoothing mechanism is changed. This tightness result for RS-Abn, together with the empirical results in Figure 2, suggests that RS-Del produces certificates which are tight or close to tight in practice, at least for the Hamming distance threat model. This is an interesting observation, since it is unclear how to derive a tight, computationally tractable certificate for RS-Del.

### E.3 Summary

Our evaluation shows that RS-Del provides non-trivial robustness guarantees with a low impact on accuracy. The certified radii we observe are close to the best radii theoretically achievable using our mechanism. For the Levenshtein byte-level edit distance threat model, we obtain radii of a few hundred bytes in size, which can certifiably defend against attacks that edit headers of PE files [37, 38, 52]. However, certifying robustness against more powerful attacks that modify thousands or millions of bytes remains an open challenge. By varying the detection threshold, we show that certification can be performed asymmetrically for benign and malicious instances. This can boost the certified radii of malicious instances by a factor of 4 in some cases. While there are no prior methods to use as baselines for the Levenshtein distance threat model, our comparisons with RS-Abn [24] for

the Hamming distance threat model show that RS-Del outperforms RS-Abn in terms of both accuracy and robustness.

# F  Evaluation of robustness to published attacks

In this appendix, we empirically evaluate the robustness of RS-Del to several published evasion attacks. By doing so, we aim to provide a more complete picture of robustness, as our certificates are conservative and may *underestimate* robustness to real attacks, which are subject to additional constraints (e.g., maintaining executability, preserving a malicious payload, etc.). We introduce the attacks in Appendix F.1, provide details of the experimental setup in Appendix F.2 and discuss the results in Appendix F.3.

## F.1  Attacks covered

Table 9: Evasion attacks used in our evaluation. The *attack distance* refers to the median Levenshtein distance computed on a set of 500 attacked files from the Sleipnir2 test set. We use a closed source implementation of *Disp* and open source implementations of the remaining attacks based on `secml-malware` [93].

| Attack | Supported settings | Attack distance | Optimizer | Description |
|--------|--------------------|-----------------|-----------|-------------|
| *Disp* [17] | White-box, black-box | 17.2 KB | Gradient-guided | Disassembles the PE file and displaces chunks of code to a new section, replacing the original code with semantic nops. |
| *Slack* [20] | White-box | 34.7 KB | Fast Gradient Sign Method [2] | Replaces non-functional bytes in slack regions or the overlay of the PE file with adversarially-crafted noise. |
| *HDOS* [37] | White-box, black-box | 58.0 B | Genetic algorithm | Manipulates bytes in the DOS header of the PE file which are not used in modern Windows. |
| *HField* [38] | White-box, black-box | 17.0 B | Genetic algorithm | Manipulates fields in the header of the PE file (debug information, section names, checksum, etc.) which do not impact functionality. |
| *GAMMA* [31] | Black-box | 2.10 MB | Genetic algorithm | Appends sections extracted from benign files to the end of a malicious PE file and modifies the header accordingly. |

We consider five recently published attacks designed for evading static PE malware detectors as summarized in Table 9. The attacks cover a variety of edit distance magnitudes from tens of bytes to millions of bytes. While attacks that edit millions of bytes arguably fall outside our edit distance-constrained threat model, we include one such attack (*GAMMA*) to test the limits of our methodology. We note that four of the five attacks are able to operate in a black-box setting and can therefore be applied directly to RS-Del. However, the white-box attacks are designed for neural network malware detectors with a specific architecture. In particular, they assume the network receives a raw byte sequence as input, that the initial layer is an embedding layer, and that gradients can be computed with respect to the output of the embedding layer. Although these architectural assumptions are satisfied by the base MalConv model, they are not satisfied by RS-Del, because additional operations are applied before the embedding layer and the aggregation of base model predictions is not differentiable. In Appendix H, we adapt the white-box attacks for RS-Del by applying two tricks: (1) we apply the smoothing mechanism *after* the embedding layer, and (2) we replace majority voting with soft aggregation following Salman et al. [94].

Table 10: Success rates of direct attacks against RS-Del and the NS baseline. A lower success rate is better from our perspective as a defender, as it means the model is more robust to the attack. The model that achieves the lowest success rate for each attack/dataset is highlighted in boldface.

| | | | Attack success rate (%) | |
|---|---|---|---|---|
| Setting | Attack | Dataset | NS | RS-Del |
| White-box | *Disp* [17] | Sleipnir2 | 73.8 | **56.7** |
| | | VTFeed | 94.1 | **74.5** |
| | *Slack* [20] | Sleipnir2 | **57.9** | 85.3 |
| | | VTFeed | 96.0 | **43.9** |
| Black-box | *HDOS* [37] | Sleipnir2 | 0.0 | 0.0 |
| | | VTFeed | 0.0 | 0.0 |
| | *HField* [38] | Sleipnir2 | 0.607 | **0.0** |
| | | VTFeed | 0.990 | **0.0** |
| | *Disp* [17] | Sleipnir2 | 0.809 | **0.0** |
| | | VTFeed | 10.9 | **0.0** |
| | *GAMMA* [31] | Sleipnir2 | 99.2 | **54.1** |
| | | VTFeed | **76.2** | 100.0 |

## F.2 Experimental setup

Since some of the attacks take hours to run for a single file, we use smaller evaluation sets containing malware subsampled from the test sets in Table 3. The evaluation set we use for Sleipnir2 consists of 500 files, and the one for VTFeed consists of 100 files (matching [17]). We note that our evaluation sets are comparable in size to prior work [18, 20, 51]. For each evaluation set, we report attack success rates against malware detectors trained on the same dataset.

Since all attacks employ greedy optimization with randomization, they may fail on some runs, but succeed on others. We therefore repeat each attack 5 times per file and use the best performing attacked file in our evaluation. We define the attack success rate as the proportion of files initially detected as malicious for which at least one of the 5 attack repeats is successful at evading detection. Lower attack success rates correlate with improved robustness against attacks. We permit all attacks to run for up to 200 attack iterations of the internal optimizer. Early stopping is enabled for those attacks that support it (*Disp*, *Slack*, *GAMMA*), which means the attack terminates as soon as the model's prediction flips from malicious to benign.

Where possible, we run *direct attacks* against RS-Del and compare success rates against NS as a baseline. We also consider *transfer attacks* from NS to RS-Del as an important variation to the threat model, where an attacker has limited access to the target RS-Del during attack optimization. When running direct attacks against RS-Del, we use a reduced number of Monte Carlo samples ($n_{\text{pred}} = 100$) to make the computational cost of the attacks more manageable. For both direct and transfer attacks against RS-Del, we set $p_{\text{del}} = 97\%$ and perform deletion and certification at the byte-level (BYTE).

## F.3 Results

The results for direct attacks against RS-Del are presented in Table 10. For both the Sleipnir2 and VTFeed datasets, we observe that the robustness of RS-Del is superior (or equal) to NS against four of the six attacks. The two cases where RS-Del's robustness drops compared to NS are for the strongest attacks: *Slack* and *GAMMA*. The results for transfer attacks from NS to RS-Del are presented in Table 11. Almost all of the attacks transfer poorly to RS-Del. In most cases the attack success rates drop to zero or single digit percentages. We hypothesize that *Slack* and *GAMMA* make such drastic changes to the original binary that they can overwhelm the malicious signal—enough to cross the decision boundary—akin to a good word attack [95]. We find that *HDOS* and *HField* are ineffective for both RS-Del and the baseline NS. Both attacks change up to 58 bytes in the header, and tend to fall within our certifications.

Table 11: Success rates of attacks transferred from NS to RS-Del.

| | | | Attack success rate (%) | |
|---|---|---|---|---|
| Setting | Attack | Dataset | NS | RS-Del |
| White-box | *Disp* [17] | Sleipnir2 | 73.8 | 0.414 |
| | | VTFeed | 94.1 | 0.0 |
| | *Slack* [20] | Sleipnir2 | 57.9 | 2.90 |
| | | VTFeed | 96.0 | 1.01 |
| Black-box | *HDOS* [37] | Sleipnir2 | 0.0 | 0.0 |
| | | VTFeed | 0.0 | 0.0 |
| | *HField* [38] | Sleipnir2 | 0.607 | 0.0 |
| | | VTFeed | 0.990 | 0.0 |
| | *Disp* [17] | Sleipnir2 | 0.607 | 0.0 |
| | | VTFeed | 10.9 | 0.0 |
| | *GAMMA* [31] | Sleipnir2 | 99.2 | 99.6 |
| | | VTFeed | 76.2 | 100.0 |

## G    Efficiency of RS-Del

In this appendix, we discuss the training and computational efficiency of RS-Del. We provide comparisons with RS-Abn [24], which serves as a baseline in Appendix E.2 for a more restricted Hamming distance threat model.

**Computational efficiency**    Table 12 reports wall clock times for training and prediction. For training, we measure the time taken to complete 1 epoch of stochastic gradient descent on the Sleipnir2 training set, where inputs are perturbed by the smoothing mechanism. For prediction, we measure the time taken for a single 1MB input file using $n_{\mathrm{pred}} = 1000$ Monte Carlo samples. We split the prediction time into two components: (1) the time taken to generate perturbed inputs from the smoothing mechanism and (2) the time taken to aggregate predictions for the perturbed inputs using the base MalConv model. All times are recorded on a desktop PC fitted with an AMD Ryzen 7 5800X CPU and an NVIDIA RTX3090 GPU, using our PyTorch implementation of RS-Del and RS-Abn. We execute training and prediction for the base model on the GPU, and the smoothing mechanism on the CPU. We use a single PyTorch process, noting that times may be improved by running the smoothing mechanism in parallel or on the GPU.

We now make some observations about the results. First, we note that training is an order of magnitude faster for RS-Del compared with RS-Abn. We attribute this speed-up to the deletion smoothing mechanism of RS-Del, which drastically reduces the dimensionality of inputs, thereby reducing the time taken to perform forward and backward passes for the base model. On the contrary, the ablation smoothing mechanism of RS-Abn does not alter the dimensionality of inputs, so it does not have a performance advantage in this respect. Second, we observe that the total prediction time for RS-Del is approximately 150% faster than for RS-Abn. We expect this difference is also due to the effect of dimensionality reduction for the deletion smoothing mechanism.

**Training efficiency**    Training curves for the base MalConv models used in RS-Del and RS-Abn are provided in Figure 7 for the Sleipnir2 dataset. Due to convergence issues for RS-Abn, we adapted training to incorporate gradient clipping when updating the embedding layer. This addresses imbalance in the gradients arising from the dominance of masked (ablated) values in the perturbed inputs. However, even with this fix, we observe slower convergence to a higher loss value for RS-Abn than for RS-Del. Combining the results of Table 12 and Figure 7, we conclude that RS-Abn beats RS-Abn in terms of training efficiency as it requires both fewer epochs to converge and takes less time per epoch.

Table 12: Comparison of runtime efficiency for two models: RS-Del (our method with byte-level deletion) and RS-Abn [24]. The first column of wall times measures the time taken to train each model for one epoch on Sleipnir2. The second and third columns of wall times measure the time taken to make a prediction for a 1MB input file. This is split into two components: the time taken to generate $n_{\text{pred}} = 1000$ perturbed inputs from the smoothing mechanism (second column) and the time taken to pass the perturbed inputs through the base model (third column).

| | | | Wall time (s) | |
| | | | Predict | |
| Model | Parameters | Train 1 epoch | Smoothing | Base model |
| --- | --- | --- | --- | --- |
| RS-Del | $p_{\text{del}} = 90\%$ | 354 | 10.42 | 0.070 |
| RS-Abn [24] | $p_{\text{ab}} = 90\%$ | 1692 | 15.29 | 0.352 |
| RS-Del | $p_{\text{del}} = 99\%$ | 329 | 8.79 | 0.043 |
| RS-Abn [24] | $p_{\text{ab}} = 99\%$ | 1788 | 15.60 | 0.352 |

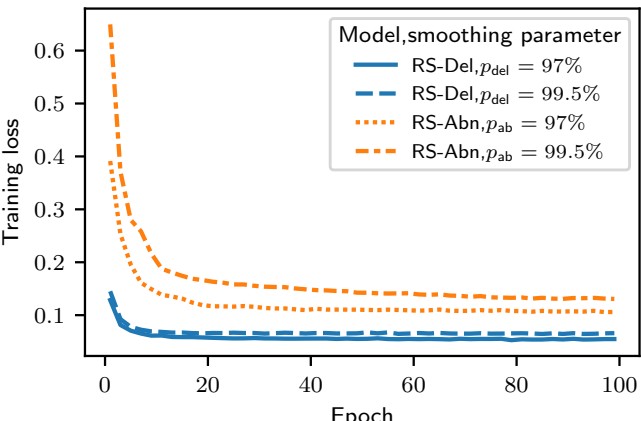

Figure 7: Training curves for RS-Del (our method with byte-level deletion) and RS-Abn [24] for the Sleipnir2 dataset.

## H  Adapting attacks for smoothed classifiers

In this appendix, we show how to adapt gradient-guided white-box attacks to account for randomized smoothing.

We consider a generic family of white-box attacks that operate on neural network-based classifiers, where the first layer of the network is an embedding layer. Mathematically, we assume the classifier under attack $f$ can be decomposed as

$$f = f_{\text{embed}} \circ f_{\text{soft}} \circ f_{\text{pred}} \tag{23}$$

where

- $f_{\text{embed}}$ is the embedding layer, which maps an input sequence $\boldsymbol{x} \in \mathcal{X}$ of length $n = |\boldsymbol{x}|$ to an $n \times d$ array of $d$-dimensional embedding vectors;

- $f_{\text{soft}}$ represents the subsequent layers in the network, which map an $n \times d$ embedding array to a probability distribution over classes $\mathcal{Y}$; and

- $f_{\text{pred}}$ is an optional final layer, which maps a probability distribution over classes to a prediction (e.g., by taking the $\arg\max$ or applying a threshold).

The attack may query any of the above components in isolation and compute gradients of $f_{\text{soft}}$. For instance, the attacks we consider [20, 37, 17] optimize the input in the embedding space $e \in \mathbb{R}^{n \times d}$, by computing the gradient $\frac{\partial f_{\text{soft}}(e)}{\partial e}$.

While the above setup covers attacks against MalConv, or classifiers with similar architectures, it is not directly compatible with smoothed classifiers. There are two incompatibilities. First, a direct implementation of a smoothed classifier (as described in Section 3.1 and Figure 1) does not decompose like (23). And second, the aggregation of hard predictions from the base classifiers is non-differentiable. We explain how to address these incompatibilities below.

**Leveraging commutativity** Consider a smoothed classifier composed from a base classifier $h$ and smoothing mechanism $\phi$, and suppose the base classifier decomposes as in (23). Following Section 3.1 and Figure 1, the smoothed classifier's confidence score for class $y$ can be expressed as

$$p_y(\boldsymbol{x}) = \text{smooth}_N(\boldsymbol{x}; \phi, h_{\text{embed}} \circ h_{\text{soft}} \circ h_{\text{pred}} \circ \mathbf{1}_{\square=y}) \tag{24}$$

where

$$\text{smooth}_N(\boldsymbol{x}; \phi, f) = \frac{1}{N} \sum_{i=1}^{N} f(\boldsymbol{z}_i), \quad \text{with } \boldsymbol{z}_i \sim \phi(\boldsymbol{x})$$

is the (empirical) smoothing operation. This expression does not immediately decompose like (23), because the embedding layer is applied to the perturbed input $\phi(\boldsymbol{x})$, not $\boldsymbol{x}$. Fortunately, we can manipulate the expression into the desired form by swapping the order of $\phi$ and $h_{\text{embed}}$. To do so, we extend the definition of $\phi$ to operate on an embedding array so that $h_{\text{embed}}(\phi(\boldsymbol{x})) = \phi(h_{\text{embed}}(\boldsymbol{x}))$, i.e., $\phi$ and $h_{\text{embed}}$ commute. In particular, this can be done for randomized deletion (RS-Del) by applying the deletion edits to embedding vectors along the first dimension. Then (24) can equivalently be expressed as

$$p_y(\boldsymbol{x}) = \underbrace{h_{\text{embed}}}_{f_{\text{embed}}} \circ \underbrace{\text{smooth}_N(\phi, h_{\text{soft}} \circ h_{\text{pred}} \circ \mathbf{1}_{\square=y})}_{f_{\text{soft}}}(\boldsymbol{x}). \tag{25}$$

**Soft aggregation** While (25) decomposes as required, the $f_{\text{soft}}$ component is not differentiable. This is due to the presence of $h_{\text{pred}}$, which is an $\arg\max$ layer. To proceed, we replace the aggregation of predictions by the aggregation of softmax scores as proposed by Salman et al. [94]. This yields a differentiable approximation of the smoothed classifier:

$$p_y(\boldsymbol{x}) \approx \underbrace{h_{\text{embed}}}_{f_{\text{embed}}} \circ \underbrace{\text{smooth}_N(\phi, h_{\text{soft}} \circ \square_y)}_{f_{\text{soft}}}(\boldsymbol{x}).$$

Salman et al. note that this approximation performs well, and is empirically more effective than an alternative approach proposed by Cohen et al. [14, Appendix G.3].

# I  Review of randomized ablation

In this appendix, we review *randomized ablation* [24], which serves as a baseline for the Hamming distance threat model in our experiments. It is based on *randomized smoothing* (see Section 3.1) like our method, however the smoothing mechanism and robustness certificate differ. In this review, we formulate randomized ablation for sequence classifiers; we refer readers to Levine and Feizi [24] for a formulation for image classifiers.

## I.1  Ablation smoothing mechanism

Randomized ablation employs a smoothing mechanism that replaces a random subset of the input elements with a special null value NA. Levine and Feizi use an encoding for the null value tailored for images, that involves doubling the number of channels. This encoding is not suitable for discrete sequences, so we instead augment the sequence domain $\Omega$ with a special null value: $\Omega \rightarrow \Omega \cup \{\text{NA}\}$. The hyperparameter controlling the strength of ablation must also be adapted for our setting. Levine and Feizi use a hyperparameter $k$ that corresponds to the number of elements *retained* in the output. This is ineffective for inputs that vary in length, so we scale $k$ in proportion with the input length. Specifically, we introduce an alternative hyperparameter $p_{\text{ab}} \in (0, 1)$ that represents the fraction of ablated elements and set $k(|\boldsymbol{x}|) = \lceil (1 - p_{\text{ab}})|\boldsymbol{x}| \rceil$.

Mathematically, the ablation mechanism has the following distribution when applied to an input sequence $\boldsymbol{x}$:

$$\Pr[\phi(\boldsymbol{x}) = \boldsymbol{z}] = \sum_{\epsilon \in \mathcal{E}_{k(|\boldsymbol{x}|)}(\boldsymbol{x})} \frac{1}{\binom{|\boldsymbol{x}|}{k(|\boldsymbol{x}|)}} \mathbf{1}_{\text{ablate}(\boldsymbol{x},\epsilon)=\boldsymbol{z}},$$

where $\mathcal{E}_k = \{\epsilon \in \mathcal{E}(\boldsymbol{x}) : |\epsilon| = k\}$ consists of all sets of element indices of size $k$ and

$$z_i = \text{ablate}(\boldsymbol{x}, \epsilon)_i = \begin{cases} x_i, & \text{if } i \in \epsilon \text{ "retained"}, \\ \texttt{NA}, & \text{if } i \notin \epsilon \text{ "ablated"}. \end{cases}$$

*Remark* 10. Hyperparameter $p_{\text{ab}}$ has a similar interpretation as $p_{\text{del}}$ for randomized deletion, in that both hyperparameters control the proportion of sequence elements hidden (by ablation or deletion) from the base classifier.

### I.2 Hamming distance robustness certificate

Randomized ablation provides a Hamming distance ($\ell_0$) certificate that guarantees robustness under a bounded number of arbitrary substitutions. Levine and Feizi provide two certificates: one that makes use of the confidence score for the predicted class, and another that makes use of the top two confidence scores. We present the first certificate here, since it matches the certificate we consider in Section 4 and it is the simplest choice for binary classifiers (the focus of our experiments).

To facilitate comparison with randomized deletion, we reuse notation and definitions from Section 4. Recall from (15) that $\tilde{\rho}(\boldsymbol{x}; \mu_y)$ is a lower bound on the smoothed classifier's confidence score $p_y(\bar{\boldsymbol{x}}; h)$ that holds for any input $\bar{\boldsymbol{x}}$ in the edit distance ball of radius $r$ centered on $\boldsymbol{x}$ and any base classifier $h$ such that $p_y(\boldsymbol{x}; h) = \mu_y$. For randomized ablation, we can replace the edit distance ball with a Hamming distance ball and obtain the following result (see [24] for the derivation):

$$\tilde{\rho}(\boldsymbol{x}; \mu_y) = \mu_y - 1 + \frac{\binom{|\boldsymbol{x}|-r}{k(|\boldsymbol{x}|)}}{\binom{|\boldsymbol{x}|}{k(|\boldsymbol{x}|)}}.$$

Recall from Proposition 3, that the smoothed classifier is certifiably robust if $\tilde{\rho}(\boldsymbol{x}; \mu_y) \geq \nu_y(\boldsymbol{\eta})$, where $\boldsymbol{\eta}$ denotes the smoothed classifier's tunable decision thresholds. Hence the certified radius $r^\star$ is the maximum value of $r \in \{0, 1, 2, \ldots\}$ that satisfies

$$\mu_y - 1 + \frac{\binom{|\boldsymbol{x}|-r}{k(|\boldsymbol{x}|)}}{\binom{|\boldsymbol{x}|}{k(|\boldsymbol{x}|)}} - \nu_y(\boldsymbol{\eta}) \geq 0.$$

This maximization problem does not have an analytic solution, however it can be solved efficiently using binary search since the LHS of the above inequality is a non-increasing function of $r$. In practice, the exact confidence score $\mu_y$ can be replaced with a $1 - \alpha$ lower confidence bound $\underline{\mu}_y$ to yield a probabilistic certificate that holds with probability $1 - \alpha$ (see procedure in Figure 1 and Corollary 8).

Since randomized ablation and randomized deletion both admit Hamming distance certificates, it is interesting to compare them. The following result, first proved by Scholten et al. [96, Proposition 5] for a related mechanism, shows that randomized deletion admits a tighter certificate.

**Proposition 11.** *Consider a randomized deletion classifier (RS-Del) with deletion probability $p_{\text{del}} = p$ and a randomized ablation classifier (RS-Abn) with ablation fraction $p_{\text{ab}} = p$. Suppose both classifiers make the same prediction $y$ with the same confidence score $\mu_y$ for input $\boldsymbol{x}$ and that we are interested in issuing a Hamming distance certificate of radius $r$. Then the lower bound on the confidence score over the certified region is tighter for RS-Del than for RS-Abn.*

*Proof.* From Theorem 7 we have

$$\tilde{\rho}_{\mathsf{RS\text{-}Del}}(\boldsymbol{x}; \mu_y) = \mu_y - 1 + p^r$$

$$= \mu_y - 1 + \left(\frac{|\boldsymbol{x}| - (1-p)|\boldsymbol{x}|}{|\boldsymbol{x}|}\right)^r$$

$$\geq \mu_y - 1 + \left(\frac{|\boldsymbol{x}| - \lceil(1-p)|\boldsymbol{x}|\rceil}{|\boldsymbol{x}|}\right)^r$$

$$= \mu_y - 1 + \prod_{j=1}^{r} \frac{|\boldsymbol{x}| - k(|\boldsymbol{x}|)}{|\boldsymbol{x}|}$$

$$\geq \mu_y - 1 + \prod_{j=1}^{r} \frac{(|\boldsymbol{x}| - k(|\boldsymbol{x}|) - j + 1)}{(|\boldsymbol{x}| - j + 1)}$$

$$= \mu_y - 1 + \frac{\binom{|\boldsymbol{x}| - r}{k(|\boldsymbol{x}|)}}{\binom{|\boldsymbol{x}|}{k(|\boldsymbol{x}|)}}$$

$$= \tilde{\rho}_{\mathsf{RS\text{-}Abn}}(\boldsymbol{x}; \mu_y)$$

$\square$

*Remark* 12. Jia et al. [25] extend randomized ablation to top-$k$ prediction, while at the same time proposing enhancements to the Hamming distance certificate. These enhancements also apply to regular classification and involve: (1) discretizing lower/upper bounds on the confidence scores; and (2) using a better statistical method to estimate lower/upper bounds on the confidence scores. However, both of these enhancements would have a negligible impact in our experiments, as we shall now explain. The first enhancement tightens lower/upper bounds on the confidence scores by rounding up/down to the nearest integer multiple of $q :- 1/\binom{|\boldsymbol{x}|}{k(|\boldsymbol{x}|)}$. This improves the lower/upper bound by at most

$$q \leq \min\left\{\left(\frac{|\boldsymbol{x}| - k}{|\boldsymbol{x}|}\right)^{|\boldsymbol{x}| - k}, \left(\frac{k}{|\boldsymbol{x}|}\right)^k\right\} \leq \frac{1}{|\boldsymbol{x}|}$$

if the number of retained elements satisfies $1 \leq k \leq |\boldsymbol{x}|$, as is the case in our experiments. However, since we consider inputs of length $|\boldsymbol{x}| \geq 10^3$, this means the improvement in the bound due to discretization is at most $q \leq 10^{-3}$. This improvement is comparable to the resolution of our estimator $(1/n_{\mathrm{bnd}} \sim 10^{-4})$, and consequently, discretization will not have a discernible impact. The second enhancement involves simultaneously estimating lower/upper bounds using a statistical method called SimuEM [97]. SimuEM has been demonstrated to improve tightness when there are multiple classes [25], however it has no impact when there are two classes (as is the case in our experiments).

