# OpenReview forum: "RS-Del: Edit Distance Robustness Certificates for Sequence Classifiers via Randomized Deletion"
_NeurIPS.cc/2023/Conference — NeurIPS 2023 poster_

### Official Review · Reviewer_MXYp · 2023-06-17

**Soundness:** 2 fair
**Presentation:** 3 good
**Contribution:** 3 good
**Rating:** 4
**Confidence:** 4

**Summary:**

This paper proposes **RS-Del** -- a novel certified defense that provides guarantees w.r.t. insertion, deletion, and substitution of bytes within a variable length input.  As its name indicates, RS-Del is based on randomized smoothing.  However, unlike classic randomized smoothing, the authors cannot certify robustness using the Neyman-Pearson lemma and instead propose a novel certification scheme.

**Strengths:**

The paper has a lot to like.  Overall the writing quality was good. Ideas were structured and explained well.  The writing was clear.

Certified and robust training literature is dominated by works centered on some variant of an $\ell_p$ threat model. It is rare to see a paper that prevents as novel of a certified threat model as this one.  Reading a paper with such an innovative threat model is refreshing and welcome.
* I would further argue that some in the adversarial ML community do not appreciate how myopically the research community focuses on narrow definitions of robustness.  The paper's "Related Work" section does a good job of explaining this point.  I appreciated reading the author's perspective in this regard.

The most closely related work I am aware of is Saha et al. (2023) (which the authors cite).  I find RS-Del's threat model and type of guarantee much more useful and innovative.  Having looked at both papers, I find this work substantially stronger and more compelling.

**Weaknesses:**

There are a lot of things I like about the paper.  That notwithstanding, this paper feels more appropriate for a security venue than an ML venue.  Understanding malware and the meaningfulness of guaranteeing robustness up to $m$ bytes is generally beyond the expertise of most ML readers (and reviewers).  I believe this paper would be a better fit and better appreciated at a venue like S&P or Usenix than at NeurIPS.  The misalignment with the venue slightly reduced my score.

### Empirical Evaluation

To the extent of my knowledge, no other method provides guarantees w.r.t. to Levenshtein distance.  The authors select randomized ablation (RA) [Levine and Feizi 2020] as the primary baseline for Hamming distance.  Levine and Feizi's RA is far from the state-of-the-art $\ell_0$ certified method at this point.  Two works consistently outperform vanilla RA:

[1] Jinyuan Jia, Binghui Wang, Xiaoyu Cao, Hongbin Liu, and Neil Zhenqiang Gong. "Almost Tight $\ell_0$-norm Certified Robustness of Top-k Predictions against Adversarial Perturbations". ICLR 2022. https://openreview.net/forum?id=gJLEXy3ySpu.

[2] Zayd Hammoudeh and Daniel Lowd. "Feature Partition Aggregation: A Fast Certified Defense Against a Union of Sparse Adversarial Attacks". arXiv 2302.11628. https://arxiv.org/abs/2302.11628

Jia et al. is based on RA and provides tight(er) certification analysis.  Hammoudeh and Lowd use model ensembles.  This paper does not cite either of these papers (Hammoudeh and Lowd is a newer preprint so I understand why the authors may not have seen it). However, at minimum, Jia et al.'s version of RA must be a baseline of comparison.
* I would have voted "Borderline Accept" with the paper as is had at least Jia et al. (2022) been used as a baseline. Updating the baselines is a necessary condition for me to increase my score. I cannot recommend accepting this paper without at least Jia et al. (2022) as a baseline.  I strongly recommend adding both baselines or at minimum a compelling explanation in the rebuttal why one is not possible.

### Interpreting the Empirical Results

The authors specify their certified guarantees in terms of the number of bytes.  I have two primary concerns about this framing.

1. Existing certified methods also specify their guarantees in absolute terms. For example, RA provides median guarantees on CIFAR10 of 7 pixels. Providing guarantees in absolute terms makes sense when instances have a fixed size; its trivial to convert that result into a relative quantity (e.g., 1% of pixels). When test instances have variable sizes as in the case here, no simple conversion exists. For a reader to be able to appreciate how meaningful a guarantee of 128 bytes is, we need to understand the typical range of malware sizes.  At an ML venue, such information needs to be provided in the main paper.  Moreover, it is the author(s)'s responsibility to explain how the size of the guarantee changes with malware size.

2. Providing certified guarantees as a fraction of malware size is only a small part of the equation. I expect that most ML readers do not know a priori the extent of changes that must be made to a program to induce a change of 128 bytes. For example, would simply recompiling the program with no code changes and just compiler setting changes be sufficient to change the binary by 128 bytes?  I do not have expertise in that area, and the authors do not educate the reader in this regard.  I consider this choice to be a significant oversight.

### Interpreting the Empirical Results

Certified $\ell_0$ method randomized ablation (RA) is used as the primary Hamming distance baseline.  However, the authors provide little to no explanation of how RA works.  I think this is important so a reader not familiar with RA can appreciate what is being tested in the experiments. Moreover, the authors need to explain how their method differs from RA when restricted to exclusively the substitution setting.


**Questions:**

See my comments above under "Weaknesses".

**Limitations:**

I understand why the authors placed Table 1 where they did (put simply -- space).  However, the table's explanation does not appear until the next page.  When I first saw the table, I did not understand the table and thought perhaps I had missed something.  I think separating the tables on separate pages would be a better choice.  To prevent a significant increase in length, NeurIPS allows text to wrap around inline figures and tables.

The authors could provide more intuitions about why their setting needs turnable decision thresholds $\eta_y$ while most other randomized smoothing applications do not.  Tying this explanation to their case study/expected end use would make the explanation much more compelling.

---

> ### Author Rebuttal · Authors · 2023-08-10
>
> We thank the reviewer for their detailed and constructive feedback.
>
> ### RE: fit for NeurIPS
> We believe our paper is aligned for NeurIPS (see below), and note that this was not a concern raised by other reviewers. Our paper advances certified robustness for generic sequence classifiers, providing both algorithmic and theoretical contributions for a previously unexplored threat model. These are contributions the machine learning community has historically been interested in—by our count 10 papers on certified robustness appeared at NeurIPS last year. We view malware as a compelling domain to test our method, given growing reliance of malware detection methods on machine learning and the domain's real threat of evasion attacks. Our empirical evaluation focuses on certification not malware defense.
>
> ### RE: baselines for Hamming distance
> Thank you for suggesting Jia et al. (2022) and Hammoudeh & Lowd (2023) as alternative baselines and for offering to raise your score. We discuss them separately below.
>
> **Jia et al. (2022):**
> After receiving this review, we implemented Jia et al.'s method and incorporated it into the certified accuracy plot (updated Figure 2 in the **rebuttal PDF**). We find the certified accuracy curve for Jia et al. (denoted RS-Jia) is identical to the curve for the existing baseline by Levine & Feizi (denoted RS-Abn). This is not surprising given Jia et al.'s certification method for top-k prediction is very similar to Levine & Feizi's method when instantiated for binary classification ($k = 1$). More specifically, Jia et al. use the same ablation smoothing mechanism, and while they do derive a tighter certificate, the difference is negligible in a high dimensional setting. To explain why the difference is negligible, we note that Jia et al.'s tighter certificate comes from a tighter lower bound on the smoothed classifier's confidence score. In particular, they tighten the lower bound by rounding up to the nearest integer multiple of $q = 1 / {d \choose r}$ (where $d$ is the length of the sequence and $r = \lceil p_\mathrm{abn} d \rceil$ is the number of ablated elements). The difference between the original lower bound and the tighter one is no larger than $q \leq 10^{-3}$ in our experiments, since $d \geq 10^{3}$ for the sequences we consider and $p_\mathrm{abs} \leq 0.999$. Thus in the worst case, the difference is of a similar order to the resolution of the Monte Carlo estimator ($1/4000 = 2.4 \times 10^{-4}$), which explains why the tighter bound does not have a discernible impact.
>
> **Hammoudeh & Lowd (2023):**
> This work looks interesting as it is not based on randomized smoothing like the other baselines for Hamming distance. We will add a citation to it, however adding it as a baseline would require non-trivial changes for two reasons.
> First, the formulation is fundamentally different: Hammoudeh & Lowd assume all inputs have the same dimensionality (Section 2 of their paper), whereas the dimensionality of our inputs vary. Second, the current code base would require significant refactoring to work with large datasets, as it loads the entire training set and test set in memory. This is infeasible for our datasets which are hundreds of GB in size.
>
> ### RE: absolute versus relative certified radius
> Our certificates are independent of sequence length (see Table 1). This means the _absolute_ certified radius is better aligned with the theory—it allows us to compare radii without artificially introducing a dependence on sequence length. That being said, we agree that _relative_ certified radius is useful as a complementary metric. In our original submission, we reported the median relative certified radius (referred to as the normalized certified radius) in Table 7 of Appendix E.1. In our revision, we have updated Figures 4 and 5 to include a plot of certified accuracy versus relative radius, alongside the existing plot for absolute radius (see **rebuttal PDF**). We will endeavour to include such results for relative radius in the main paper.
>
> ### RE: interpretation of edit distance
> Whether an edit distance can be regarded as "small" or "large" depends on the application and threat model. For our application to malware, we report the median edit distance induced by several attacks in Table 9. The distance varies from 10's of bytes to millions of bytes, depending on the attack. We appreciate and agree with the reviewer's comments on the need to assist the reader; we will include a discussion of interpreting edit distances in malware in terms of cited/tested attacks, file sizes, and with reference to the relative certified radii from the previous point.
>
> ### RE: background on randomized ablation
> Thank you for this suggestion. We will add a brief description of randomized ablation (RS-Abn) in Section 5, noting the following differences from RS-Del: (1) RS-Abn performs substitution with a special "masking" value whereas RS-Del performs deletion; (2) the number of elements to edit is fixed for RS-Abn but follows a binomial distribution for RS-Del; (3) RS-Abn provides a Hamming distance certificate, whereas RS-Del provides a generalized edit distance certificate. We will also provide a more detailed summary of RS-Abn in a new appendix, where we will also provide an explicit comparison with RS-Del for the Hamming distance setting.

---

> > ### Comment · Reviewer_MXYp · 2023-08-11
> > **Partial Reply**
> >
> > I do not have time today to provide a full reply, but I wanted to provide some partial feedback in the expectation of writing more in the coming days.  The authors are most welcome to provide a response to this partial feedback in the meantime.
> >
> > > Fit at NeurIPS
> >
> > I agree that certified methods have a clear place at NeurIPS; there was never a question of that.  However, this paper certifies what I consider is a particularly specialized type of certified robustness targeted primarily at one application -- malware (the paper even contains a malware case study).  That's not necessarily a bad thing; as I said in my review, "the paper has a lot to like."
> >
> > Let's restrict discussion here to the malware case, which I claim is the paper's strongest motivation and biggest focus.   Intuitively, your method provides some robustness guarantee $r$.  I could be wrong, but I find it unlikely that the vast majority of folks in the NeurIPS community possess the a priori knowledge to assess whether RS-Del's certified guarantees are meaningful in the malware space; such an assessment requires nitty-gritty knowledge of compiled binaries, which I don't expect most to know.  I gave an example in my review, where only the malware's compiler settings are changed with no code changes (other examples are possible where just the code is refactored or reordered). In such cases, will a malware's compiled binary, in general, change by more than your median $r$?  I admittedly do not know, but I lean towards the total changes far exceeding RS-Del's guaranteed $r$ (I could be wrong).  Perhaps the other reviewers know the answer to this question, with me the odd one out, but I would be surprised.  You may disagree, but I think it is a fair general rule that if most reviewers and relevant readers cannot assess the semantic meaningfulness of your empirical results, there is some misalignment with the venue.
> >
> > There are ways to mitigate what I contend is (limited) venue misalignment.  If the vast majority of the venue's target audience lacks critical background knowledge, the paper needs to provide it (it's not a perfect solution but better than alternatives).  Your rebuttal promises that "*we will include a discussion of interpreting edit distances in malware in terms of cited/tested attacks, file sizes, and with reference to the relative certified radii from the previous point.*" I think that's great.  Still, I hope your reply can provide significant specificity here on those details so, at minimum, I can use that information in my assessment of the paper and its empirical results.
> >
> > > Jia et al. (2022) Baseline
> >
> > Thank you for adding this baseline.  I would have a priori expected a (much) larger gap.  Would it be possible to share the updated source code for verification?  I would like to run the experiment and verify the implementation of Jia et al.'s version, including the hyperparameters.  Of course, I affirm the code will not be used for anything other than reviewing and will be deleted at the end of the review period. I believe authors can provide anonymized links to the ACs, who can then share it with the reviewers.
> >
> > > Hammoudeh & Lowd (2023):
> >
> > Thank you for looking into this. Your explanation makes sense given the limited rebuttal time.

---

> > > ### Author Response · Authors · 2023-08-15
> > >
> > > ### Interpreting edit distance
> > > We can certify edit distance radii up to 128 bytes without significant loss in accuracy. This corresponds to relative radii 0–9% (as a fraction of file size). Our certificates cover real attacks against malware detectors: attacks in Demetrio et al. (2019) and Nisi et al. (2021) change up to 58 bytes in a file's header, and Lucas et al. (2021, 2023) change as little as 1% of a file. These attacks make iterative  localized edits (e.g., editing individual bytes or machine instructions) and are therefore well aligned with the edit distance threat model. Other attacks (e.g., Demetrio et al., 2021) make larger changes in edit distance, outside the threat model.
> > >
> > > This is no different from other domains where certification is applied, such as vision, where adopted threat models (e.g., $\ell_p$ perturbations) cover a limited set of possible attacks.
> > >
> > > For further perspective on edit distance, we can consider rule-based classification tools widely used for malware analysis in industry. For example YARA is a rule-based tool supported by VirusTotal.
> > > Running Nextron System's public YARA rule set on a sample of binaries from VTFeed, we find 83% of matching rules are sensitive to fewer than 128 bytes. This implies most rules can be evaded by perturbations covered by our certified radii, further demonstrating relevance. This is unsurprising given manual rules are typically sensitive to a small byte patterns (function names, file paths, keys, urls, etc.).
> > >
> > > **References**
> > > * Lucas et al. "Adversarial Training for Raw-Binary Malware Classifiers." USENIX Security '23.
> > > * Demetrio et al. "Explaining Vulnerabilities of Deep Learning to Adversarial Malware Binaries." ITASEC '19.
> > > * Nisi et al. "Lost in the Loader: The Many Faces of the Windows PE File Format." RAID '21.
> > > * Lucas et al. "Malware Makeover: Breaking ML-Based Static Analysis by Modifying Executable Bytes." AsiaCCS '21.
> > > * Demetrio et al. "Functionality-Preserving Black-Box Optimization of Adversarial Windows Malware." In IEEE Transactions on Information Forensics and Security 16 (2021), 3469–3478
> > >
> > > ### Jia et al. (2022) baseline
> > > We are happy to share our code and we've asked for permission from the AC/SAC (thanks for your separate reply on that request).
> > >
> > > As an alternative, we would like to further explain why the baselines by Jia et al. and Levine & Feizi perform almost identically in our setting with _high dimensional inputs and binary classes_. In our previous response, we focused on the binary setting, however the empirical improvements reported by Jia et al. hold for the multiclass setting. There's a key difference between these settings as we'll now explain.
> > >
> > > Consider the statistical estimation of the lower/upper bounds on the classifier's confidence scores. Jia et al. estimate both bounds simultaneously using a method called SimuEM proposed in their ICLR'20 paper. SimuEM yields a tighter estimate than Levine & Feizi's estimate when there are multiple classes, however _the estimates are equivalent when there are only two classes_ (as is the case in our experiments). This means the only difference between Jia et al. and Levine & Feizi in the binary setting is rounding of lower/upper bounds to integer multiples of $q = 1 / {d \choose r}$. We showed in our previous response that rounding has a negligible impact: it tightens the bound by no more than $10^{-3}$ (but sometimes much less than this) when $d \geq 10^3$.
> > >
> > > If the reviewer wishes to interpret Jia et al.'s empirical results in our binary setting, then the above analysis shows that _only the effect of rounding is relevant, not the effect of SimuEM (since it reduces to Levine & Feizi's method for two classes)_. Fortunately, Jia et al. report an ablation study in Tables 1 & 2 of their paper that isolates the impact of rounding and SimuEM. Specifically, they consider the following combinations:
> > >
> > > | Label in Tables 1 & 2 | Statistical estimation method | Rounding lower/upper bound |
> > > |--|--|--|
> > > | Levine & Feizi (2019) | Clopper-Pearson for top score | No |
> > > | Levine & Feizi (2019) + SimuEM (Jia et al. 2020) | SimuEM | No |
> > > | "Our method" (referring to Jia et al. 2022) | SimuEM | Yes |
> > >
> > > To examine the effect of rounding only (without SimuEM) we can compare the rows labeled "Levine & Feizi (2019) + SimuEM (Jia et al. 2020)" and "Our method". **The certified accuracies reported in these rows are identical for CIFAR-10 and ImageNet at all levels of ablation ($r$)**. This suggests that rounding to an integer multiple of $q$ has no discernible benefit, and that the improved performance of Jia et al. is due to SimuEM.
> > >
> > > In summary, we have demonstrated that Jia et al.'s results are consistent with our findings in a high-dimensional binary setting. We only expect Jia et al.'s method to outperform Levine & Feizi's in a multiclass setting where SimuEM makes a difference.
> > >
> > > We thank the reviewer for previously offering to increase their score following our evaluation of Jia et al.

---

### Official Review · Reviewer_bHHQ · 2023-07-04

**Soundness:** 4 excellent
**Presentation:** 4 excellent
**Contribution:** 4 excellent
**Rating:** 9
**Confidence:** 5

**Summary:**

This paper tackles the issue of applying randomized smoothing to discrete sequences under the Levenshtein edit distance. Because the underlying sequence is discrete, it necessitates new mathematical approaches to proving the robustness. Enticingly, the edit distance is bounded by employing only the delete operation, making implementation and application far easier than would have otherwise been possible. Noting the inequity in attack risk for malware/security applications, effective bias terms $\eta$ are added to allow asymmetric certificates of robustness. Since no prior work has obtained this result, they compare against a prior hamming certificate, and show significant improvement over prior results in the smaller Hamming space.

The application to malware is very apt, but this is also relevant to all the domains of which the Levenshtein edit distance is applicable: genomics, time series analysis, epidemiology, linguistics/NLP, etc.

**Strengths:**

1. This is allowing results in a whole new category of model/hypothesis spaces for randomized smoothing.
2. The results show significant improvement as the maximum radius is reached over closest-prior alternatives
3. The method is easy to implement
4. Real-world consideration to asymmetric attack profile is considered and incorporated into the approach, while not firefighting the multi-class support.
5. The paper is beautifully written and the best I have read in reviewing over probably the past 2 years.

**Weaknesses:**

1. The only weakness to the paper is some missing references. The asymmetric nature of benign vs malicious attacks was previously noted in "Non-Negative Networks Against Adversarial Attacks" and "Adversarially Robust Malware Detection Using Monotonic Classification", and the two should be cited accordingly. Notably, both only deal with the additive threat model $O = [ \texttt{ins} ]$, filling in the broader contribution of this work in completing the Levenshtine operation set.
2. If I had to add a second quibble, I would ask the authors to caution the reader about applying the % certification to other datasets, because each dataset will differ and industrial-scale classification datasets are beyond the scope of proving the mechanism's correctness. (I could see industry research getting weird reviews from academics who don't understand this when attempting to publish case-studies).

**Questions:**

I honestly can't think of any questions to ask beyond random directions of obviously future work. The appendix is detailed, has code, and the paper is exceptionally well written.

To the AC, please note that my shorter review isn't a remark against thoroughness in reading the work. It just answered all the questions I would have asked. This work is of exceptional quality, presentation, and impact.

**Limitations:**

This work has no realistic limitations beyond not "solving" all problems in one go. Other challenges of randomized smoothing naturaly remain, and can't be held against the authors given the work's goals and how well it has achieved them.

---

> ### Author Rebuttal · Authors · 2023-08-10
>
> We thank the reviewer for providing such encouraging feedback. We are pleased the reviewer recognized the novelty of our work in extending randomized smoothing to new threat models for discrete domains. We also appreciate the reviewer's pragmatic assessment of the limitations.
>
> ### W1: Missing references
> Thank you for identifying references on the asymmetry of misclassification costs for malware detection by Incer et al. (2018) and Fleshman  et al. (2018). We have added citations to both references in Remark 1. While we are not the first to point out this asymmetry, to the best of our knowledge we are the first to consider asymmetric certified robustness for classification.
>
> ### W2: Caution about applying certification results to other datasets
> Thanks for making the point that experiments cannot prove the correctness of the robustness guarantee (which is covered by the analysis in Section 4). While the experiments support the effectiveness of our approach, we understand the reviewer's concern that readers may falsely assume that the results generalize to other datasets. In our revision, we have identified this as a limitation in the final paragraph of Section 7.

---

> > ### Comment · Reviewer_bHHQ · 2023-08-10
> >
> > >asymmetric certified robustness for classification.
> >
> > https://arxiv.org/abs/2302.01961
> >
> > I believe this pre-print has the same goal, but is in a very different space of models. I don't think being first is relevant to your contribution here. That we can obtain Levenshtein robust bounds using only deletion is highly surprising.
> >
> > All my concerns are satisfied. I leave my score as-is. You really should go talk with some bioinformatics folks about these results, there are a lot of potential inferences one could make in gene expression classification and potentially verifiably identifiable interactions by bounding the edit distance. At least, that's what my friends in such places mention as a challenge toward their work.

---

> > > ### Author Response · Authors · 2023-08-11
> > >
> > > > https://arxiv.org/abs/2302.01961
> > >
> > > Thanks for the pointer to this preprint. We will cite it as related work on asymmetric certification, noting that it covers a different family of classifiers (Lipschitz feature map composed with a convex function) and a different threat model ($\ell_p$). We agree with the reviewer that it's not related to our primary contribution—extending certification for the edit distance threat model.
> > >
> > > Thanks also for suggesting applications in bioinformatics. We agree that it seems to be a natural fit for our work and certainly worthwhile exploring.

---

### Official Review · Reviewer_xRpL · 2023-07-07

**Soundness:** 2 fair
**Presentation:** 4 excellent
**Contribution:** 3 good
**Rating:** 6
**Confidence:** 4

**Summary:**

The paper proposes a general randomized smoothing approach for certifying robustness concerning arbitrary perturbations defined in Levenshtein distance. The critical challenges of proposing the randomized smoothing approach are
  1) how to design the smoothing distribution? The paper uses a deletion distribution by randomly removing tokens in the inputs.
  2) how to derive the bound? The paper extensively derives a loose bound using the Neyman-Pearson lemma on the proposed deletion distribution.

The approach is effective on malware detection datasets. The paper also proposes an interesting tuning mechanism to favor false positives over false negatives, as in the malware detection scenario, the escaping of malware is a bigger problem than false alarms on benign software.

**Strengths:**

1. The paper proposes a general randomized smoothing approach for certifying robustness concerning arbitrary perturbations defined in Levenshtein distance. The approach is effective on malware detection datasets.
2. The paper also proposes an interesting tuning mechanism to favor false positives over false negatives, as in the malware detection scenario, the escaping of malware is a bigger problem than false alarms on benign software.
3. The paper also evaluates RS-Del to empirical attacks.

**Weaknesses:**

1. It is astonishing to see the classifier maintains a high certified accuracy when >90% of the inputs are deleted. I don't think it is possible to do so in NLP datasets, e.g., movie reviews datasets like SST2 and IMDB. Deleting >90% of the movie reviews definitely destroy the meaning of the reviews.
2. missing related works:
    * In lines 40-42, the paper states "there is also no work for ... along with substitution or additive perturbation". This is not the case. ARC [1] is a deterministic approach for certifying robustness of LSTMs given arbitrary perturbation spaces, including insertion, deletion, substitution, and their combinations. However, ARC can only be used for LSTMs and the certified radius is much smaller compared to RS-Del. ARC also scales linearly with respect to the certified radius while RS-Del's time complexity is constant, e.g, equal to the number of sample in Monte Carlo sampling. ARC may introduce more over-approximation (e.g. looser bound) than RS-Del due to the interval bound propagation (IBP) used in ARC.
    * Masking [3,4] has been used as a randomized smoothing approach for certifying the robustness of NLP models with respect to word substitutions.
3. In Table 10, RS-Del does not perform better than NS on Slack-VTFeed and GAMMA-Sleipnir2. So it is "four out of six" instead of "five out of six". However, the NS baseline is quite weak, how about comparing RS-Del to other empirical defense approaches for malware detection?

**Questions:**

Comments:
1. $m$ never appears outside Lemma 4. Either removing $m$ from Lemma 4 or providing intuitive relation between $m$ and $p_{del}^{|\bar{x}|-|x|}$.
2. In line 286, Table 7->Table 8.

Questions:
1. Can we further improve the bound following Lee et al. [2]? In [2], the bound is to first prove that $\forall h \in \mathcal{F}(x, \mu_y)$ the minimal value of $p_y(\bar{x};h)$ will always be achieved when $dist(\bar{x},x)=r$, e.g., when the attacker tries to utilize the attack budget as much as possible. Then they can exactly compute Eq (13) without approximation. The case in [2] is easier since they only allow substitutions, but in this paper deletions and insertions are also allowed. However, in the proof of Theorem 7, the minimizer is achieved when $n_{ins}=n_{del}=0$, indicating that the possibility of tightening the bound as [2].
2. Is Corollary 6 a looser bound, e.g, LHS $\le$ RHS instead of LHS = RHS? The proof seems to assume the substituted ones and inserted ones won't be counted in the LCS, but they potentially can be.
3. In line 45, the paper states "we consider input sequences of bounded and varying length". What's "unbounded length"? It seems all inputs have bounded length.
4. What's the performance of $f_b$?
5. However, the NS baseline is quite weak, how about comparing RS-Del to other empirical defense approaches for malware detection?

**Limitations:**

The paper addresses some of the limitations. For other limitations, please refer to my points in Weakness and Questions.


[1] Certified Robustness to Programmable Transformations in LSTMs. Yuhao Zhang, Aws Albarghouthi, and Loris D'Antoni

[2] Guang-He Lee, Yang Yuan, Shiyu Chang, and Tommi Jaakkola. Tight Certificates of Adversarial Robustness for Randomly Smoothed Classifiers.

[3] Certified Robustness to Text Adversarial Attacks by Randomized [MASK]. Jiehang Zeng, Xiaoqing Zheng, Jianhan Xu, Linyang Li, Liping Yuan, Xuanjing Huang

[4] Randomized Smoothing with Masked Inference for Adversarially Robust Text Classifications. Han Cheol Moon, Shafiq Joty, Ruochen Zhao, Megh Thakkar, Xu Chi

---

> ### Author Rebuttal · Authors · 2023-08-10
>
> We thank the reviewer for thoroughly engaging with our work and providing detailed feedback.
>
> ### W1: High deletion >90% doesn't harm accuracy
> We offer an explanation in Appendix E.1 (lines 922-929). In short, it's important to realize that a deletion probability of 90% does not mean 90% of the sequence elements are inaccessible to RS-Del. Rather, 90% of the sequence elements (on average) are inaccessible for a given Monte Carlo sample. When RS-Del is run using 4000 Monte Carlo samples, each sequence element is accessed $4000 \times (1 - 0.9) = 40$ times in expectation.
>
> ### W2: Related work
> Thank you for drawing our attention to these papers. In our revision, we have cited Zhang et al. (2021) as a rare example of work that goes beyond the substitution threat model. However, we agree with the reviewer's characterization of its limitations. In particular, the perturbation spaces that can be certified in practice are quite limited (e.g., deletion of up to 2 stop words). We have also added citations to Zeng et al. (2023) and Moon et al. (2023) as examples of work covering the synonym substitution threat model (complementing refs [28] and [80] in our paper).
>
> ### W3: Interpretation of Table 10
> Our claim that RS-Del achieves the lowest attack success rate (ASR) for "five out of six attacks" holds for each dataset individually. In our revision, we have adopted the reviewer's suggestion and only count a "win" for RS-Del if it achieves the lowest ASR on _both_ datasets.
>
> ### C1: Removing $m$ from Lemma 4
> We think it's important to introduce the bijection $m$ in Lemma 4, as it's used immediately after Lemma 4 to rewrite terms that appear in the smoothed classifier's confidence score. In particular, $m$ specifies which edits to $\mathbf{\bar{x}}$ can be expressed in terms of edits to $\mathbf{x}$ without changing the summand $s$ (up to a proportionality constant).
>
> ### C2: Incorrect reference
> Thanks, we have fixed this.
>
> ### Q1: Improving the bound following Lee et al. (2019)
> Lee et al. provide a generic framework for computing robustness certificates for randomized smoothing based on the Neyman-Pearson lemma. We previously considered their framework and determined it would be computationally infeasible for our mechanism/threat model. To explain why, consider the first step in their framework: computing a pointwise certificate that guarantees the smoothed classifier's prediction at $\mathbf{x}$ does not change at a neighboring input $\mathbf{\bar{x}}$. This requires partitioning the support of the deletion mechanism into regions, such that the relative likelihood of perturbing $\mathbf{x}$ and $\mathbf{\bar{x}}$ to any point in the region is constant. For our mechanism, the relative likelihood does not simplify in general and takes $O(2^{\max \{|\mathbf{x}|, |\mathbf{\bar{x}}|\}})$ time to evaluate in the worst case. This makes the first step of their framework infeasible, let alone the second step, where one must search for the worst-case pointwise certificate in the edit-distance ball.
>
> ### Q2: Inequality or equality in Corollary 6
> We believe we’ve found the source of confusion. The expression for the lower bound $\tilde{\rho}(\mathbf{x}, \mathbf{\bar{x}}, \mu_y)$ in eqn (14) conceals a dependence on the LCS ($\mathbf{z}^\star$). In Corollary 6, we instantiate $\tilde{\rho}(\mathbf{x}, \mathbf{\bar{x}}, \mu_y)$ for a _particular_ LCS to obtain a loose lower bound on $\rho(\mathbf{\bar{x}}, \mathbf{x}, \mu_y)$ (this is fine since the bound holds for _any_ LCS). The LCS we choose is defined in terms of the cost-minimizing edit path from $\mathbf{\bar{x}}$ to $\mathbf{x}$, consisting of $n_\mathrm{sub}$ substitutions, $n_\mathrm{del}$ deletions and $n_\mathrm{ins}$ insertions. Specifically, it is the sequence obtained from $\mathbf{x}$ by deleting the $n_\mathrm{sub}$ elements substituted in $\mathbf{x}$ and deleting the $n_\mathrm{ins}$ elements inserted in $\mathbf{x}$ (see Appendix B.4). This means the equality in Corollary 6 is correct, provided one understands that the LHS depends on the LCS. To avoid confusion, we have made the dependence on the LCS ($\mathbf{z}^\star$) explicit (in our revision) by using $\tilde{\rho}(\mathbf{\bar{x}}, \mathbf{x}, \mu_y, \mathbf{z}^\star)$ in eqn (14) and propagating the change through eqn (15) and Corollary 6.
>
> ### Q3: Meaning of “unbounded length” sequence
> We used this phrase to emphasize that our method does not place any limits on sequence length—e.g., we do not require that sequences are padded to a common length. However, upon reflection, we think "variable length" is sufficient to capture this meaning. We have therefore removed "unbounded length" in our revision.
>
> ### Q4: Performance of the base classifier
> In this work (and prior work on randomized smoothing) the base classifier is not trained to perform well as a standalone classifier. Rather, it is trained to function as a component of the smoothed classifier, where it encounters inputs transformed by the smoothing mechanism. It would therefore be unusual to conduct a standalone evaluation of the base classifier on natural inputs.
>
> ### Q5: Comparing with other empirical defenses for malware detection
> While the malware literature would benefit from a comparison of empirical defenses, we feel it would best be presented in a separate paper. The focus of this paper is on advancing certification for sequence classifiers; we have developed algorithmic innovations with our randomized deletion mechanism, and corresponding theory to prove certifications. This explains the focus on certification in our experiments, where we include Levine & Feizi (2020) (and Jia et al. (2022) in our revision) as the closest certified baselines we are aware of. The purpose of the empirical robustness experiments is not to compare against empirical defenses, but rather to explore the magnitude of common attacks (in edit distance) and assess robustness beyond radii we can certify.

---

> > ### Comment · Reviewer_xRpL · 2023-08-12
> > **Response to authors**
> >
> > For W1, I understand the discussion in Appendix E.1, but the discussion does not make sense if it is not supported by Q4. "each sequence element is accessed about 40 times in expectation", intuitively, if only 40 results out of 4000 are correct, then it is still not a majority of them, making $p_A$ (percentage of predictions of the correct label) very small. So I still think the discussion is not convincing.
> >
> > Overall, I think that this paper should be accepted.

---

> > > ### Author Response · Authors · 2023-08-19
> > >
> > > Thank you for the helpful discussion. If there's more needing to be shared still, for either the reviewer/AC, please let us know.

---

### Official Review · Reviewer_r6FS · 2023-07-16

**Soundness:** 3 good
**Presentation:** 3 good
**Contribution:** 2 fair
**Rating:** 6
**Confidence:** 3

**Summary:**

This paper aimed to design a certified defense for discrete sequence classifiers against edit distance-bounded adversaries. This method exploited randomized smoothing mechanism to consturct the defense and proposed RS-Del to confer robustness against adversarial delection, insertion and substitution edits.

**Strengths:**

1. Different from most prior work, this paper focused on protecting models with discrete inputs (e.g., binary executables, source codes and PDF files), which was interesting and was meaningful in the real world.

2. The instructions for the proposed methodology were relatively clear (including the explanations of some theorems and lemmas).

**Weaknesses:**

1. Although the form of the input data was different, the defenseive mechanisms for continuous fixed-dimensional inputs (such as the methods in lines 31 to 35) may also work, but this did not seem to be adequately presented. Could the authors conduct some discussion or even comparative experiments?

2. The abbreviations needed to be supplemented with the full name when they appear for the first time (e.g., RS-Del).

**Questions:**

Please see weaknesses.

**Limitations:**

The authors have stated the limitations of their work.

---

> ### Author Rebuttal · Authors · 2023-08-10
>
> Thank you for reviewing our work and appreciating our contribution to certification for discrete modalities.
> We respond to specific feedback below.
>
> ### W1: Why can't defense mechanisms for continuous fixed-dimensional inputs be used
> A key reason why certified defenses for continuous fixed-dimensional inputs cannot be used, is because the certificates
> they produce are ill-defined in our setting.
> To see why, consider the most common $\ell_p$ certificate, which guarantees robustness for any additive perturbation
> $\mathbf{\delta} \in \mathbb{R}^d$ whose $\ell_p$-norm is smaller than some specified value $r$.
> If we try to apply this guarantee in our setting, it says that we can add a real-valued vector to a byte sequence
> without changing the classifier's prediction.
> However, it doesn't make sense to add a real-valued vector such as $\mathbf{\delta} = [-0.89,  0.97, -0.94,  0.70]$
> to a byte sequence such as $\mathbf{x} = [78, 7\mathrm{a}, 2\mathrm{d}, 6\mathrm{a}]$.
> Separate from this issue, is the fact that an $\ell_p$ certificate only covers inputs that are the same length,
> which is not very useful if inputs can vary in length.
> These points are partly covered in lines 89–93 of our submission; we will add the above concrete example to improve clarity.
>
> ### W2: Spelling out abbreviations when they first appear
> Thanks, we have carefully reviewed the text to address this issue.

---

> > ### Comment · Reviewer_r6FS · 2023-08-14
> > **Comment to the rebuttal**
> >
> > Dear authors,
> >
> > Thanks for your response. Your response addresses my concerns. Thus, I am willing to give weak accept score.

---

### Author Rebuttal · Authors · 2023-08-10

We thank the reviewers for their positive reception of our paper and for providing us with constructive feedback.

We would like to draw the reviewers' attention to the attached rebuttal PDF, which contains updated figures/results in response to Reviewer MXYp. For the benefit of the other reviewers, we briefly summarize the contents below:
* Figure 2 contains a new baseline for the Hamming distance threat model by Jia et al. (2022) (denoted RS-Jia). It uses a tighter certificate compared to the existing baseline by Levine & Feizi (2020) (denoted RS-Abn), however we find the improvement is negligible in our high-dimensional setting.
* Figures 4 and 5 have been expanded to include a plot of certified accuracy as a function of normalized radius (radius divided by sequence length). This was prompted by Reviewer MXYp's suggestion to report how the size of the guarantee compares to the sequence length (which varies for each input). The new plots complement the existing plots for absolute radius in Figures 4 and 5, and the existing column in Table 7 containing the median normalized certified radius. For further discussion of the normalized radius, we refer to our third response to Reviewer MXYp.

### References
* Jia, J., Wang, B., Cao, X., Liu, H., & Gong, N. Z. "Almost Tight L0-norm Certified Robustness of Top-k Predictions against Adversarial Perturbations." ICLR'22.
* Levine, A., & Feizi, S. "Robustness certificates for sparse adversarial attacks by randomized ablation." AAAI'20.

---

### Author Response · Authors · 2023-08-14
**Sharing link to anonymized code**

Dear AC/SAC,

Reviewer MXYp has asked *"Would it be possible to share the updated source code for verification?"*. We understand this is currently against the rules communicated to authors by email Fri Aug 11 under bullet *"Remember to not reveal your identity in any comment you post. Do not use links in your comments."* However, we'd be happy to comply with the reviewer's request, if the AC/SAC could confirm that this is allowed in this case? We could use for example [https://anonymous.4open.science/](https://anonymous.4open.science/) which anonymizes GitHub repositories.

Thanks for your consideration.

---

> ### Comment · Reviewer_MXYp · 2023-08-15
> **Reviewer MXYp Comment**
>
> This was my mistake.  I did not realize that posting anonymized links could potentially be an issue during the discussion period.  If code links are not allowed, I will take the results at face value and assess them accordingly.  Apologies for my ignorance regarding the author(s)'s rebuttal constraints, and I hope this did not create unnecessary and unintended stress for the authors.

---

### Decision · Program_Chairs · 2023-09-21

**Decision:**

Accept (poster)

**Comment:**

The paper introduces a new randomized deletion mechanism for smoothing and certifying black-box models on sequence data under edit distance threat models. The reviewers liked the work as it is looking at protecting models with discrete inputs; also the presentation of the work is clear. Some weaknesses in experimental results, comparison with baselines and other methods were identified that authors properly addressed in the rebuttal period.